# Leucine-rich repeat receptor-like gene screen reveals that *Nicotiana* RXEG1 regulates glycoside hydrolase 12 MAMP detection

Yan Wang [1,2], Yuanpeng Xu [1,2], Yujing Sun [1,2], Huibin Wang[1,2], Jiaming Qi[1,2], Bowen Wan [1,2], Wenwu Ye [1,2], Yachun Lin[1,2], Yuanyuan Shao[1,2], Suomeng Dong[1,2], Brett M. Tyler[3] & Yuanchao Wang [1,2]

Activation of innate immunity by membrane-localized receptors is conserved across eukaryotes. Plant genomes contain hundreds of such receptor-like genes and those encoding proteins with an extracellular leucine-rich repeat (LRR) domain represent the largest family. Here, we develop a high-throughput approach to study LRR receptor-like genes on a genome-wide scale. In total, 257 tobacco rattle virus-based constructs are generated to target 386 of the 403 identified LRR receptor-like genes in *Nicotiana benthamiana* for silencing. Using this toolkit, we identify the LRR receptor-like protein Response to XEG1 (RXEG1) that specifically recognizes the glycoside hydrolase 12 protein XEG1. RXEG1 associates with XEG1 via the LRR domain in the apoplast and forms a complex with the LRR receptor-like kinases BAK1 and SOBIR1 to transduce the XEG1-induced defense signal. Thus, this genome-wide silencing assay is demonstrated to be an efficient toolkit to pinpoint new immune receptors, which will contribute to developing durable disease resistance.

[1] Department of Plant Pathology, Nanjing Agricultural University 210095 Nanjing China. [2] Key Laboratory of Integrated Management of Crop Diseases and Pests (Ministry of Education), 210095 Nanjing, China. [3] Center for Genome Research and Biocomputing and Department of Botany and Plant Pathology, Oregon State University, Corvallis, OR 97331, USA. Yuanpeng Xu, Yujing Sun, Huibin Wang and Jiaming Qi contributed equally to this work. Correspondence and requests for materials should be addressed to Y.W. (email: wangyc@njau.edu.cn)

Environmental microbes constantly threaten plants. Over the course of evolution, plants have developed a multifaceted innate immune system to defend against potentially harmful microbes. One of the most important determinants of plant adaptation is the capacity to perceive the evolutionarily conserved signatures of microbial pathogens, namely microbe-associated molecular patterns (MAMPs), and initiate effective defense responses accordingly[1]. In general, MAMPs can be structural elements or proteins released from microbes[2].

Plants have evolved cell surface pattern-recognition receptors (PRRs) to detect MAMPs and thereby activate immune responses[3]. Plant genomes encode hundreds of potential cell surface receptors including receptor-like kinases (RLKs) and receptor-like proteins (RLPs)[4–6]. RLKs contain a ligand-binding ectodomain, a transmembrane (TM) domain, and an intracellular kinase domain, while RLPs lack any known intracellular signaling domains. Cell surface receptors have diverse ectodomains[4]. Receptor-like genes encoding proteins with an extracellular leucine-rich repeat (LRR) domain constitute the most over-represented family according to analyses of sequenced plant genomes[4–8]. Several reports demonstrated that these LRR receptor-like genes are crucial for plant adaptation and function in various physiological processes, including development, growth, and responses to abiotic and biotic stresses[9–11].

Hitherto, only a few LRR receptor-like genes have been documented encoding PRRs capable of recognizing MAMPs or receptor-like proteins (DAMPs) and function as immune receptors[12]. These include RLKs, such as the bacterial flagellin receptors FLS2[13,14] and FLS3[15], the bacterial elongation factor Tu receptor EFR, the bacterial elicitor xup25 receptor XPS1, and the DAMP receptors PEPR1 and PEPR2[16–19]. The characterized RLPs include tomato EIX2, Ve1, and Cf4, potato ELR, and *Arabidopsis thaliana* RLP1, RLP23, RLP30, and RLP42, which recognize fungal ethylene-inducing xylanase (EIX), fungal apoplastic effector Ave1 and Avr4, *Phytophthora* elicitin INF1, an unknown *Xanthomonas* MAMP, necrosis and ethylene-inducing peptide 1-like proteins (NLPs), *Sclerotinia sclerotiorum* elicitor SCFE1, and fungal endopolygalacturonases, respectively[20–27]. Interestingly, recognition of an epitope of the bacterial cold shock protein (csp22) requires two receptors including the LRR-RLP receptor CSPR and the LRR-RLK receptor CORE[28,29]. LRR receptors, such as the LRR-RLK BAK1, also function as co-receptors, by forming complexes with multiple LRR-type PRRs where they are indispensable for MAMP recognition and subsequent immune signaling[9,10]. In addition, the LRR-RLK SOBIR1 functions as an adaptor that associates with multiple LRR-RLPs to form bi-partite equivalents of LRR-RLKs[30]. Several other LRR-RLKs, such as ERECTA, are also involved in plant immunity[31]; however, their exact activation mechanisms remain unclear. In spite of this, the vast majority of this enigmatic protein family has not yet been investigated and their role in plant growth and immunity remains largely elusive.

With the growing number of sequences of microbial genomes, multiple pathogen-secreted elicitors have been identified. A number of these elicitors represent conserved microbial patterns that can be recognized by plants, resulting in the activation of defense responses[32,33]. The glycoside hydrolase 12 (GH12) protein XEG1 identified from the soybean root rot pathogen *Phytophthora sojae* is recognized in the plant apoplast as a novel MAMP[34]. XEG1 contains a GH domain that can degrade xyloglucan and β-glucan. This xyloglucanase activity was shown to be essential for XEG1-mediated *Phytophthora* virulence, but not for plant recognition[34,35]. The GH12 proteins are widely distributed across microbial taxa and many are able to trigger cell death in plants[34,36], indicating that the recognition system is evolutionarily conserved.

*Nicotiana benthamiana* is an important model plant for the study of plant biology and plant–pathogen interactions and has several advantages over other plant species[37]. *N. benthamiana* belongs to the Solanaceous plant family, with high genomic collinearity with other Solanaceous plant species such as tomato[38,39]. *N. benthamiana* is amenable to highly efficient protein expression and virus-induced gene silencing (VIGS). Moreover, multiple MAMPs, including elicitins, trigger defense responses or plant cell death in *N. benthamiana*, making it an excellent system for dissecting the recognition events.

In this study, we examine *N. benthamiana* as a model plant and develop a toolkit that allows high-throughput characterization of LRR receptor-like genes on a genome-wide scale. We explore the genome sequence of *N. benthamiana* for potential LRR receptor-like genes and perform silencing assays of the identified LRR-RLK and LRR-RLP genes using a VIGS-based approach. The silencing efficiency is evaluated and alterations in plant growth are monitored. Using this toolkit, we successfully identify the Response to XEG1 (RXEG1) protein, which is required for response to the GH12 protein XEG1 and homologs in *N. benthamiana*. Thus, this study has established an important toolkit for the characterization of plant immune receptors.

## Results

**LRR receptor-like genes and silencing assay in *N. benthamiana*.** We scanned the genome of *N. benthamiana* using both BLAST and hidden Markov model (HMM) searches to identify fragments encoding proteins with both LRR and TM domains. These LRR receptor-like genes were used as queries for a BLAST search of the genome sequence of *N. benthamiana*. The retrieved sequences were validated using the publically available transcriptome data and manually corrected as needed. In total, 86 LRR-RLPs and 317 LRR-RLKs were identified that are listed in Table 1 and Supplementary Data 1. Given the tetraploid nature of *N. benthamiana*, two copies of highly homologous LRR receptor-like genes were often identified in the genome (Supplementary Fig. 1), which were likely derived from the two ancestors of

**Table 1 LRR-RLPs and subgroup of LRR-RLKs in *N. benthamiana***

| Type | Subgroup | Total No. | No. of RD kinase | No. of non-RD kinase |
|------|----------|-----------|------------------|----------------------|
| LRR-RLK | I | 2 | 1 | 1 |
| | II | 20 | 20 | 0 |
| | III | 65 | 0 | 65 |
| | IV | 5 | 0 | 5 |
| | V | 14 | 14 | 0 |
| | VI | 13 | 0 | 13 |
| | VIIa | 9 | 0 | 9 |
| | VIIb | 10 | 0 | 10 |
| | VIII-1 | 8 | 1 | 7 |
| | VIII-2 | 10 | 9 | 1 |
| | IX | 6 | 6 | 0 |
| | Xa | 9 | 0 | 9 |
| | Xb | 18 | 18 | 0 |
| | XI | 48 | 46 | 2 |
| | XIIa | 33 | 1 | 32 |
| | XIIb | 15 | 6 | 9 |
| | XIIIa | 6 | 6 | 0 |
| | XIIIb | 5 | 5 | 0 |
| | XIV | 14 | 4 | 10 |
| | XV | 7 | 7 | 0 |
| LRR-RLP | NA[a] | 86 | NA[a] | NA[a] |

[a] *NA* not applicable

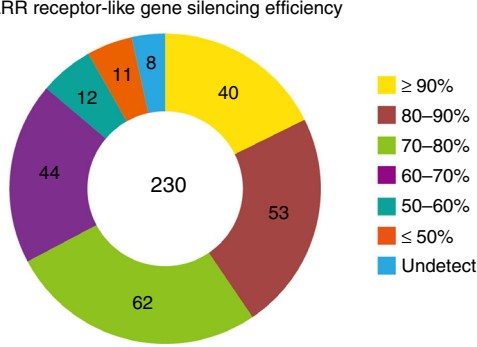

**LRR receptor-like gene silencing efficiency**

- ≥ 90%
- 80–90%
- 70–80%
- 60–70%
- 50–60%
- ≤ 50%
- Undetect

**Fig. 1** Silencing efficiency of LRR receptor-like genes in *N. benthamiana*. Expression levels were monitored by quantitative reverse transcription PCR (qRT-PCR), normalized with EF-1α, and expressed as mean fold changes relative to TRV:*GFP*-treated leaves, which was set as 1

*N. benthamiana*. Protein domain composition analyses revealed that most of the identified LRR receptor candidates contain signal peptides (SPs), LRR and TM domains, as well as a cytoplasmic tail or kinase domain, although many of these proteins are missing SPs (Supplementary Data 1). In addition, 15 LRR receptor candidates contain an extracellular malectin domain, which is a putative carbohydrate-binding domain[40]. A survey of the RD motif in the identified LRR-RLKs revealed 144 that contained an RD kinase domain and 173 that contained a non-RD kinase domain with a variety of substitutions including G, C, K, H, and S for R and N or H for D (Table 1, Supplementary Data 2). The LRR-RLKs were divided into the 20 previously delineated subfamilies[5,6] as shown in Table 1, Supplementary Fig. 1, and Supplementary Data 2. Consistent with the analysis of LRR-RLKs in 33 other Angiosperms[41], the majority of the *N. benthamiana* LRR-RLKs fall into the subgroup III and XI (Table 1). In addition, the RD kinases and non-RD kinases are separated into different subgroups.

To study the biological function of the LRR receptor-like genes, TRV silencing vectors were constructed based on the predicted receptor gene sequences. The size of the silencing fragments ranged from 144 base pairs (bp) to 494 bp (Supplementary Data 1). The silencing specificity was determined by BLAST analysis using the VIGS tool in the Sol Genomic Network (SGN) website[42]. In this case, no silencing was predicted to occur if the matching fragment was less than 21 bp. For a number of receptor-like genes with high sequence similarity, we could not design constructs to specifically target a single gene. In this case, constructs targeting more than one LRR receptor-like genes were made. In addition, several silencing constructs were made that targeted the same receptor-like genes. In total, 257 TRV constructs were generated and used for silencing assays in *N. benthamiana* (Supplementary Data 1). A survey of the silencing efficiency of 230 LRR receptor-like genes (i.e. 43 LRR-RLPs and 187 LRR-RLKs) by qRT-PCR using gene-specific primers revealed that for 199 genes the silencing efficiency was greater than 60% (Fig. 1).

For most of the LRR receptor-like genes, the silenced *N. benthamiana* plants exhibited no obvious alterations in growth and/or morphology when compared to the TRV:*GFP*-treated plants. In contrast, plants treated with the silencing construct T106, which targeted brassinosteroid-insensitive 1 (*BRI1*) homologs, were dwarfs with short stature, dark green, and wrinkled leaves (Supplementary Fig. 2a). This is consistent with the *Arabidopsis* BRI1 mutants[43]. *N. benthamiana* plants treated with the silencing constructs T123, T136, T140, or T211 repeatedly showed significantly reduced plant size or bleaching of leaves

(Supplementary Fig. 2a). Collectively, these four constructs target two genes encoding LRR-RLPs and six genes encode LRR-RLKs with less than ten extracellular LRRs. Only the two genes targeted by T136 encode an RD kinase domain while the remaining four LRR-RLK genes encode non-RD kinases with a GN or GH substitution (Supplementary Fig. 2b). Sequence similarity analysis revealed that all six LRR-RLKs were highly conserved in different plant species (Supplementary Fig. 2b). Together, these results indicate that these six previously unknown LRR-RLKs may play conserved roles in the regulation of plant growth.

**RXEG1 regulates GH12 MAMP-induced cell death.** To analyze the function of the LRR receptor-like genes in MAMP-mediated plant responses, we expressed the GH12 protein XEG1, a widely distributed MAMP in microbial taxa, in the LRR receptor-like gene-silenced *N. benthamiana* plants. XEG1-induced cell death developed in the TRV:*GFP*-treated control plants 3 days after infiltration (Fig. 2a). In contrast, cell death was significantly compromised or abolished in *N. benthamiana* leaves treated with several TRV constructs. Of the candidate TRV constructs, T2 and T33 targeted the same receptor genes and were selected for further analyses. Hereafter, the silencing constructs T2 and T33 are defined as TRV:*RXEG1-1* and TRV:*RXEG1-2*, respectively. We repeatedly detected that XEG1-induced cell death was abolished in *N. benthamiana* treated with either TRV:*RXEG1-1* or TRV:*RXEG1-2* (Fig. 2a). However, cell death induced by elicitors including INF1 and NPP1 or the membrane-localized cell death-inducing effector Avh241 was not significantly altered in TRV:*RXEG1*-treated *N. benthamiana*. Western blot analyses indicated that the expression level of each protein in TRV:*RXEG1-1* or TRV:*RXEG1-2*-treated *N. benthamiana* leaves was comparable to that with TRV:*GFP* treatment (Fig. 2a). These results indicated that the TRV:*RXEG1-1* and TRV:*RXEG1-2* targeted gene(s) encode a component essential for XEG1 recognition. Both constructs simultaneously silenced two genes encoding full length LRR-RLP receptors containing 31 LRRs in the extracellular domain and one encoding an RLP with 16 LRRs (Fig. 2b, Supplementary Fig.3). These three receptors were named as RXEG1 (*Niben101Scf03925g01010.1*), RXEGL1 (*Niben101Scf00975g01015.1*), and RXEGL2 (*Niben101Scf03925g02017.1*), respectively. To determine which LRR receptor-like gene is responsible for XEG1 recognition, we made synthesized versions of these receptor-like genes (RXEG1$^{(S)}$, RXEGL1$^{(S)}$, and RXEGL2$^{(S)}$; Supplementary Data 3) and expressed each individually with XEG1 in TRV:*RXEG1-1*- or TRV:*RXEG1-2*-treated *N. benthamiana*. All three receptors were expressed as indicated by western blotting (Fig. 2c), but could not be silenced by TRV:*RXEG1-1* or TRV:*RXEG1-2* (Supplementary Data 3). XEG1-induced cell death was only restored in TRV:*RXEG1*-treated *N. benthamiana* leaves expressing RXEG1$^{(S)}$ or a mixture of the three receptors, but not in leaves expressing the other two synthesized receptors (Fig. 2c). To be noted, expression of RXEG1$^{(S)}$ alone did not cause cell death in *N. benthamiana* (Supplementary Fig. 4). To further confirm that RXEG1 is required for XEG1 recognition, we also generated the silencing construct TRV:*RXEG1-3*, which specifically silenced RXEG1, but not the other two receptor-like genes as confirmed by qRT-PCR (Fig. 2b, Supplementary Data 3). XEG1-induced cell death was again significantly compromised in *N. benthamiana* treated with TRV:*RXEG1-3* (Fig. 2a). None of the *N. benthamiana* plants treated with each of the three *RXEG1*-silencing constructs showed significant growth alterations when compared with TRV:*GFP* treatment (Supplementary Fig. 5). Collectively, these results demonstrate that RXEG1 is essential for the XEG1-response in *N. benthamiana*.

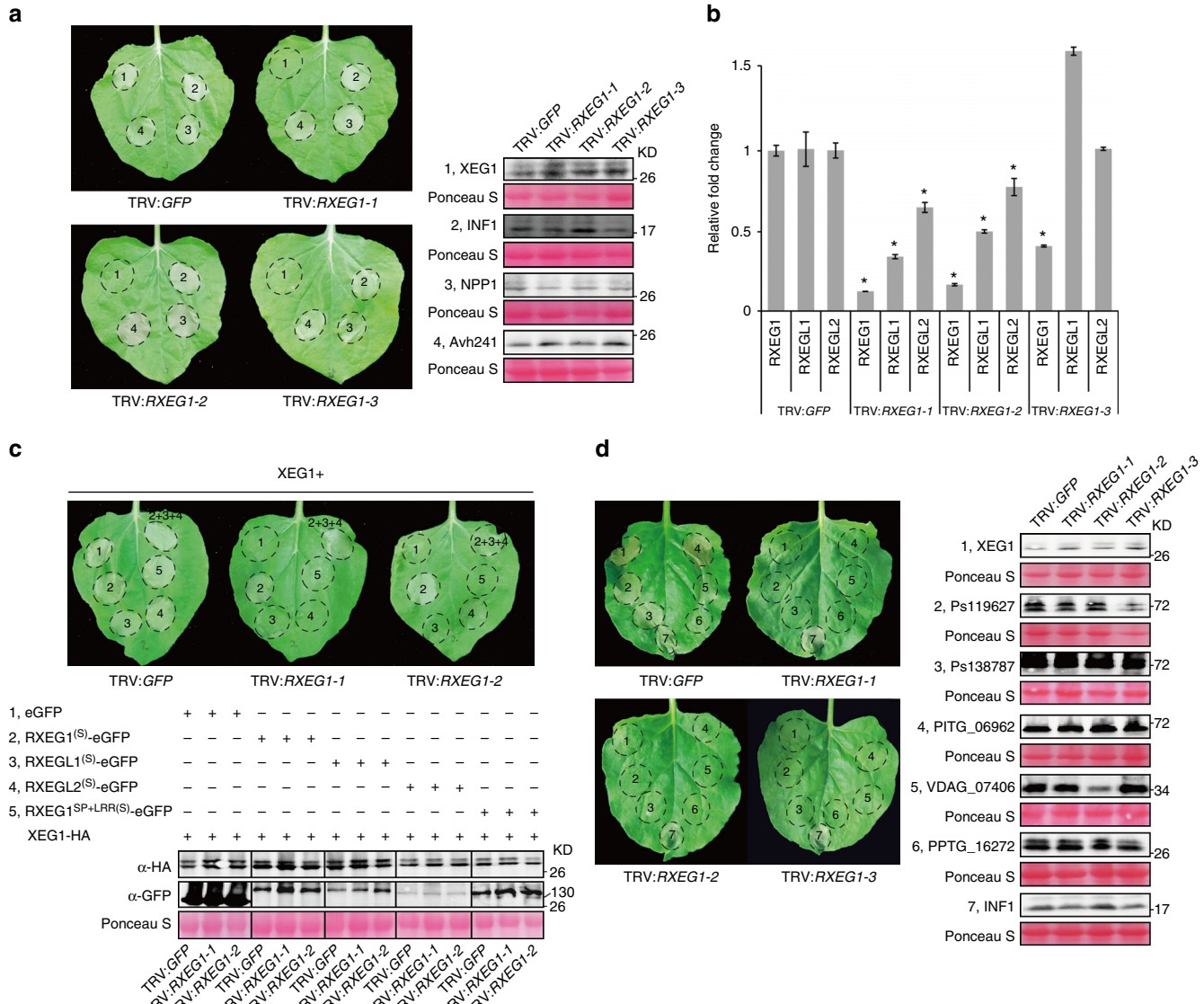

**Fig. 2** RXEG1 mediates GH12 protein recognition in *N. benthamiana*. **a** Representative leaves showing cell death induced by expression of XEG1, INF1, NPP1, or Avh241 in *N. benthamiana*. Leaves (n = 10) were photographed three days after agro-infiltration (dai). Accumulation of each protein was monitored at 2 dai with anti-HA antibody. The numbers used for western blots match the numbers on the leaves. Protein loading is indicated by Ponceau S staining. Experiments were repeated three times with similar results. **b** Relative transcript levels of RXEG1 and RXEG-likes in TRV:*GFP*- and TRV:*RXEG1*-treated *N. benthamiana*. Leaves were collected from the same position of *N. benthamiana* of similar size and used for RNA isolation. The transcript levels were determined by qRT-PCR, normalized with EF-1α, and expressed as mean fold changes [±standard error (s.e.m.)] relative to TRV:*GFP*-treated leaves, which was set as 1. *, significant differences ($P < 0.05$, Student's *t*-test). Similar results were obtained in three independent experiments. **c** Representative leaves showing expression of synthesized RXEG1 restored XEG1-induced cell death in *N. benthamiana* treated with TRV:*RXEG1-1* or TRV:*RXEG1-2*. XEG1 was co-expressed with P19 and RXEG1(S), RXEG1LRR(S), RXEGL1/2(S), or a mixture of RXEG1(S) and RXEGL1/2(S) in *N. benthamiana* leaves treated with TRV:*GFP*, TRV:*RXEG1-1*, or TRV:*RXEG1-2*. Infiltrated leaves (n = 6) was photographed at 3 dai. Total protein was isolated from infiltrated leaves at 2 dai and analyzed with anti-HA and anti-GFP antibodies. Protein loading is indicated by Ponceau S staining. Experiments were repeated three times with similar results. **d** Representative leaves showing cell death induced by expression of various GH12 proteins and INF1 in *N. benthamiana* leaves treated with TRV:*GFP* or TRV:*RXEG1*. GH12 proteins and INF1 were transiently expressed in *N. benthamiana*. Infiltrated leaves (n = 10) were photographed at 3 dai. Total protein was isolated from infiltrated leaves at 2 dai and accumulation of each protein was analyzed by western blot with anti-HA antibody. The numbers used for western blots match the numbers on the leaves. Protein loading is indicated by Ponceau S staining. Experiments were repeated three times with similar results

Since the GH12 protein family is widely distributed among different microbial taxa and multiple members can induce cell death in *N. benthamiana*[34], we next determined the role of RXEG1 in plant recognition of other GH12 proteins. We expressed GH12 proteins from various *Phytophthora* species and *Verticillium dahliae* in the TRV:*GFP* and *RXEG1*-silenced *N. benthamiana* and found cell death induced by these GH12 proteins was significantly reduced (Fig. 2d). This demonstrates

that RXEG1 mediates recognition of many different GH12 proteins.

**RXEG1 regulates XEG1-induced plant immune responses.** In addition to triggering plant cell death, XEG1 also induces immune responses including the production of ROS and the subsequent induction of the defense-related marker gene

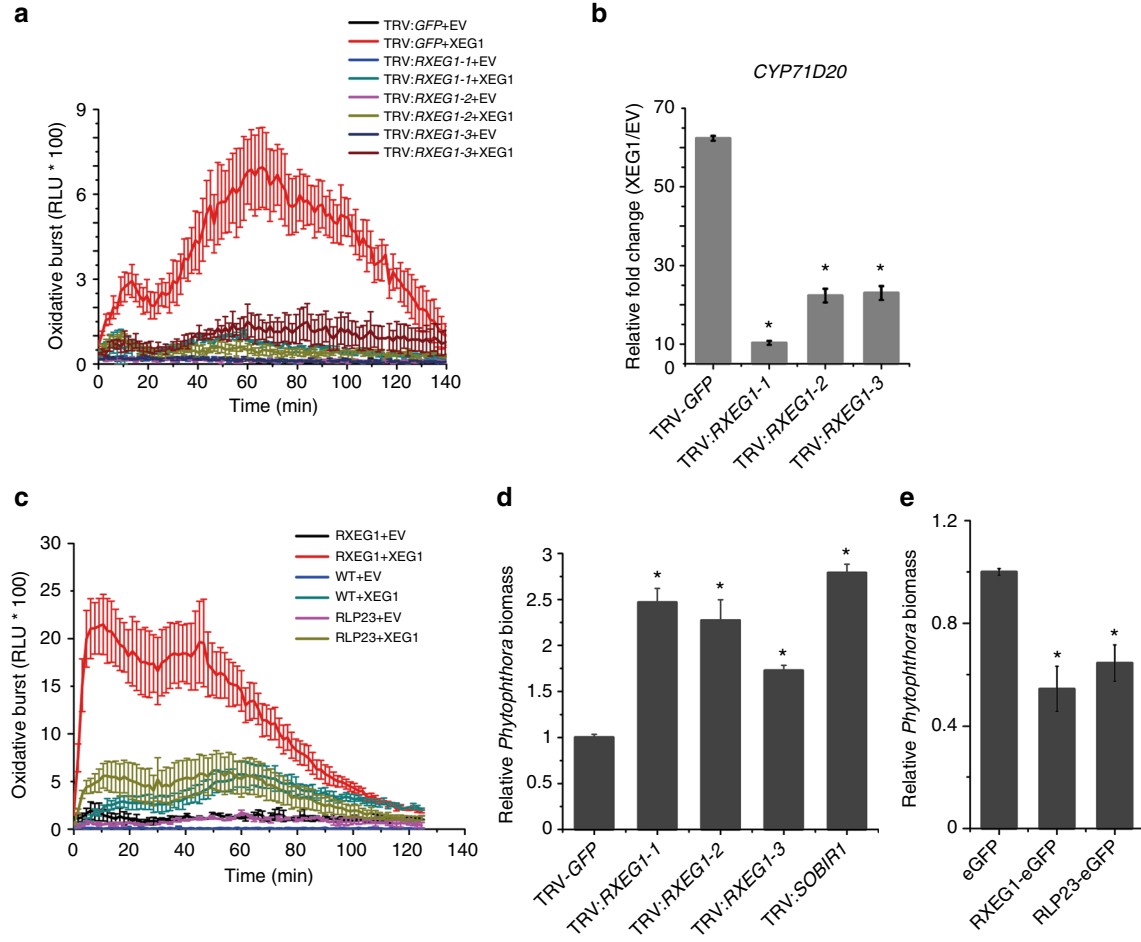

**Fig. 3** RXEG1 participates in XEG1-induced plant immunity. **a, c** Production of reactive oxygen species (ROS) in *RXEG1*-silencing (**a**) or overexpressing (**c**) *N. benthamiana* treated with 1 μM XEG1 or control (EV). Mean values (±s.e.m.) of four replicates are shown. Experiments were repeated twice with similar results. **b** Relative expression of defense-related marker gene *CYP71D20* in *RXEG1*-silencing *N. benthamiana* leaves 3 h after treatment with 200 nM XEG1 or EV. Transcript levels were quantified by qRT-PCR and normalized to EF-1α. Bars represent the mean fold changes (±s.e.m.) of the XEG1-treated leaves relative to the value in EV-treated leaves, which was set as 1. *, significant differences ($P < 0.05$, Student's *t*-test). Experiments were repeated three times with similar results. **d, e** Relative quantification of *P. parasitica* biomass using qPCR. Infected leaves ($n = 12$) were collected at 3 days post inoculation (dpi) and used for DNA isolation and qPCR analysis. *, significant differences ($P < 0.05$, Student's *t*-test). Experiments were repeated twice with similar results

*CYP71D20* (Fig. 3a, b). To verify whether RXEG1 also mediates plant defensive responses, we determined XEG1-induced ROS bursts as well as *CYP71D20* expression. In plants treated with each of the three TRV:*RXEG1* constructs, both XEG1-induced ROS production and *CYP71D20* expression were significantly compromised compared with TRV:*GFP*-treated leaves (Fig. 3a, b). However, silencing of *RXEG1* did not affect the plant defense response to the bacterial MAMP flg22 (Supplementary Fig. 6). The ability of RXEG1 to confer responsiveness to XEG1 was further confirmed by the marked increase in XEG1-induced ROS production in *N. benthamiana* leaves upon overexpression of RXEG1 (Fig. 3c). This, however, was not detected in leaves overexpressing *Arabidopsis* RLP23. These results indicate that overexpression of RXEG1 in *N. benthamiana* potentiated the plant response to XEG1.

We next evaluated whether RXEG1 plays a role in plant resistance by performing infection assays with *Phytophthora parasitica*. Leaves silencing of *SOBIR1* were inoculated and used as a positive control. Compared with leaves treated with TRV:*GFP*, *N. benthamiana* silencing of *RXEG1* or *SOBIR1* showed increased disease severity (Supplementary Fig. 7a). This difference

in disease severity was confirmed by monitoring the relative *Phytophthora* biomass with quantitative polymerase chain reaction (qPCR), where a significant increase in *Phytophthora* biomass was consistently detected in the *RXEG1*-silenced leaves (Fig. 3d). Moreover, overexpression of RXEG1 significantly reduced disease severity and *Phytophthora* biomass in *N. benthamiana* (Fig. 3e, Supplementary Fig. 7b). In this assay, *Arabidopsis* RLP23 was used as a positive control since it has been previously shown to confer resistance to *Phytophthora*[24]. Together, these results demonstrate that RXEG1 contributes to plant immunity.

**RXEG1 associates specifically to GH12 proteins in planta.** To test whether XEG1 associates with RXEG1 and the two RXEG1-like receptors, we co-expressed the C-terminal HA-tagged XEG1 with the C-terminal GFP-tagged RXEG1 or the two homologous RXEG1-like receptors in *N. benthamiana*. The XEG1 and RXEG constructs carried their normal SPs for secretion and membrane localization, respectively. As a control, we co-expressed XEG1 with the GFP-tagged *Arabidopsis* RLP23, a recently identified

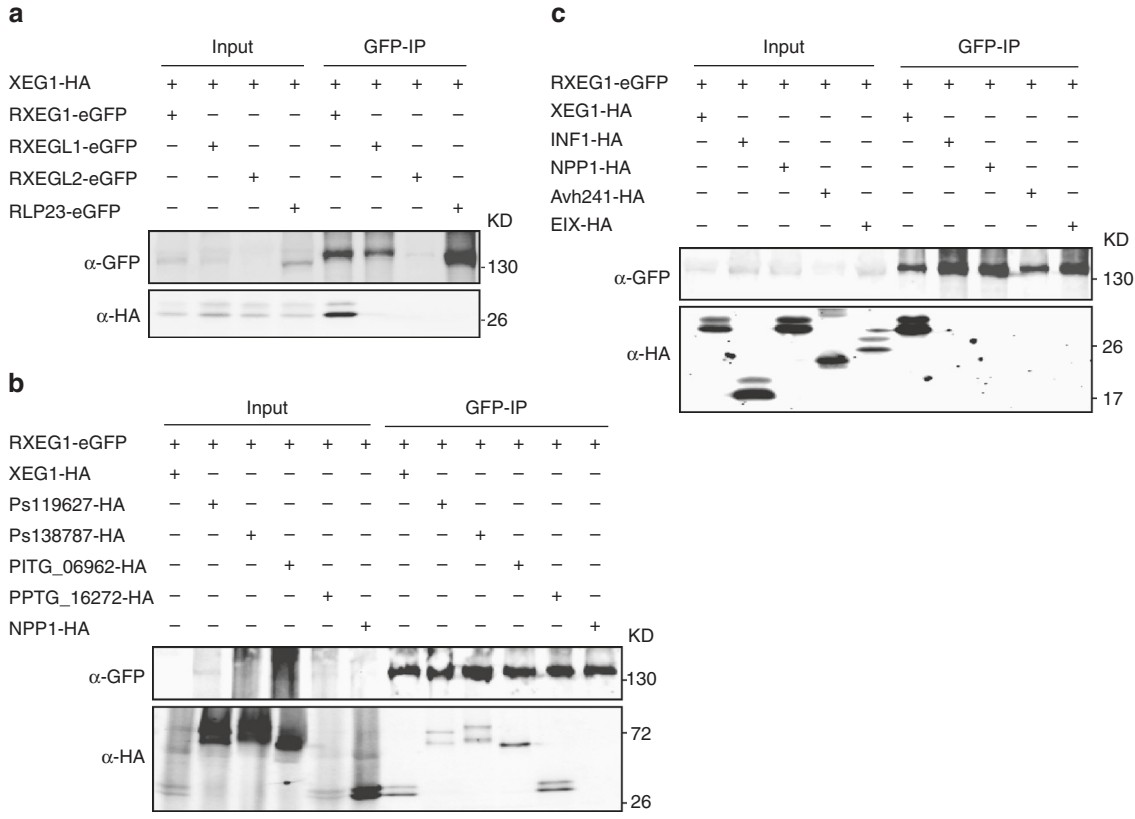

**Fig. 4** RXEG1 associates specifically with GH12 proteins in planta. **a** XEG1 associates with RXEG1, but not with RXEGL proteins or *Arabidopsis* RLP23 in planta. RLP23-GFP was used as a negative control. HA-tagged XEG1 was co-expressed with GFP-tagged RXEG1, the two RXEGL proteins, or RLP23 in *N. benthamiana*. Proteins were isolated at 2 dai, immunoprecipitated with GFP-trap® A beads, and subjected to western blotting with anti-GFP and anti-HA antibodies. **b** RXEG1 associates with different GH12 proteins in planta. NPP1 was used as a negative control. GFP-tagged RXEG1 was co-expressed with different GH12 proteins in *N. benthamiana*. Proteins were isolated at 2 dai, immunoprecipitated with GFP-trap® A beads, and subjected to western blotting with anti-GFP and anti-HA antibodies. **c** Assays for interaction between RXEG1 with XEG1 and various elicitors. GFP-tagged RXEG1 was co-expressed with HA-tagged XEG1, INF1, NPP1, Avh241, or EIX. Proteins were isolated at 2 dai, immunoprecipitated with GFP-trap® A beads, and subjected to western blotting with anti-GFP and anti-HA antibodies

nlp20 receptor in *Arabidopsis*. The association was analyzed with a co-immunoprecipitation (Co-IP) assay and western blot. As shown in Fig. 4a, we repeatedly detected specific association between XEG1 and RXEG1, but not between XEG1 and either of the RXEGL1 or RLP23. For RXEGL2, we could not conclude any association with XEG1 since the protein was poorly expressed. In addition, we checked the interactions between RXEG1 with other four different *Phytophthora* GH12 proteins using Co-IP assay and we found all the four tested GH12 proteins associate with RXEG1 in planta (Fig. 4b).

The RXEG1 orthologs in tomato, namely LeEix1 and LeEix2 (Supplementary Fig. 8), have been shown to bind the fungal Elicitor Ethylene-Inducing Xylanase (EIX), but only LeEix2 could transmit EIX signaling[21]. To further confirm the binding specificity of RXEG1 to GH12 proteins, we co-expressed GFP-tagged RXEG1 with HA-tagged XEG1, NPP1, INF1, Avh241, or EIX in *N. benthamiana*. We repeatedly detected interaction between RXEG1 and XEG1, but not between RXEG1 and the rest (Fig. 4c). We also could not detect an interaction between RXEG1 and EIX. Altogether, our results demonstrate a specific interaction between RXEG1 and GH12 proteins in planta.

**RXEG1 associates with XEG1 by the LRR domain in the apoplast.** Since RXEG1 encodes a protein with an extracellular

LRR domain (Supplementary Fig. 3), we determined whether RXEG1 associates with XEG1 via its LRR domain. We prepared constructs encoding the SP + LRR or SP + extracellular juxtamembrane (eJM) + TM + tail (ΔLRR) domain of RXEG1 (Supplementary Fig. 9), RXEGL1, or *Arabidopsis* RLP23 and co-expressed them with XEG1-HA in *N. benthamiana*. The Co-IP and western blots revealed that XEG1 co-purified with RXEG1^SP+LRR, but not with the LRR domain of RXEGL1 or RLP23 (Fig. 5a). In contrast, the ΔLRR domain of RXEG1 failed to co-purify with XEG1 (Fig. 5a). To further confirm this interaction, we isolated apoplastic fluid from *N. benthamiana* leaves co-expressing XEG1 with the LRR proteins and performed an immunoprecipitation assay. Western blot analysis showed that only the LRR domain of RXEG1 associates with XEG1 (Fig. 5b). Taken together, these results demonstrate that XEG1 can interact with RXEG1 via the extracellular LRR domain in the apoplast. To test whether the extracellular LRR domain is sufficient to mediate XEG1-induced cell death, we transiently expressed the synthesized LRR domain of RXEG1 (i.e., RXEG1^SP+LRR(S)) in the TRV:*RXEG1-1* and TRV:*RXEG1-2*-treated plants. RXEG1^SP+LRR(S) was successfully expressed in *N. benthamiana*, but failed to restore XEG1-induced plant cell death (Fig. 2c), demonstrating that the TM and intracellular tail domain are not required for XEG1 binding, but are essential for defense signal transduction.

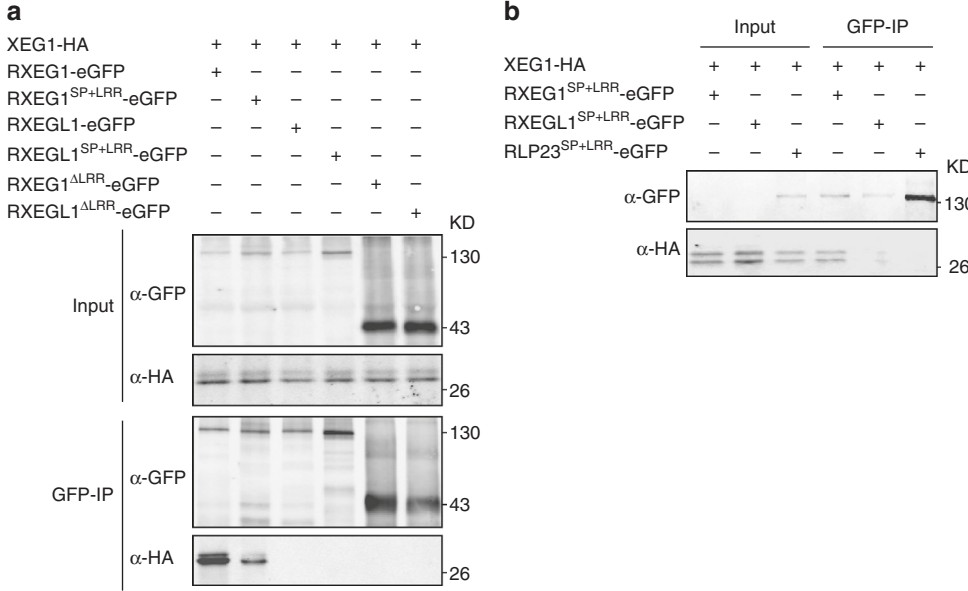

**Fig. 5** XEG1 associates with RXEG1 via the LRR domain in planta. **a** The LRR domain but not the SP + eJM + TM + tail (ΔLRR) TM domain of RXEG1 binds to XEG1 in planta. HA-tagged XEG1 was co-expressed with a GFP-tagged LRR or ΔLRR domain of RXEG1, RXEGL1, or RLP23. Protein association was analyzed by co-immunoprecipitation (Co-IP) and western blotting. **b** The LRR domain of RXEG1 binds to XEG1 in the apoplast. XEG1 was co-expressed with GFP-tagged RXEG1$^{SP+LRR}$, RXEGL1$^{SP+LRR}$, or RLP23$^{SP+LRR}$ in *N. benthamiana*. Apoplastic fluid was collected 2 dai and subjected to Co-IP and western blotting

**RXEG1 associates with BAK1 and SOBIR1 for XEG1 response.** BAK1 and SOBIR1 are two RLKs that participate in defense signaling by multiple PRRs. During screening, we found that XEG1-mediated cell death was almost abolished in TRV:*BAK1*-treated N. benthamiana, which targets the two *NbBAK1* homologs (Fig. 6a, b), consistent with a previous report[34]. In comparison, XEG1-induced cell death was only slightly compromised in *N. benthamiana* when *SOBIR1* was silenced (Fig. 6a, b). INF1-induced cell death, in contrast, was significantly reduced in TRV:*SOBIR1*-treated *N. benthamiana* (Fig. 6a). The difference in cell death was further validated with an ion leakage assay (Fig. 6c). XEG1-induced ROS production and *CYP71D20* expression were both significantly reduced in TRV:*BAK1*-treated *N. benthamiana*, and to a lesser but significant extent in leaves treated with TRV:*SOBIR1* (Fig. 6d, e). In line with this, silencing of *BAK1* or *SOBIR1* in *N. benthamiana* resulted in increased disease severity and *Phytophthora* biomass after inoculation with *P. parasitica* (Supplementary Fig. 10). These results indicate differences in the signaling pathways for XEG1-induced cell death and immune responses.

Next, we evaluated the associations between RXEG1 and the two RLKs by Co-IP. Previous reports have shown that BAK1 associated with PRRs in a ligand-dependent manner while SOBIR1 constitutively interacted with LRR-RLP PRRs even in the absence of ligand[24]. Our results for *Arabidopsis* RLP23 are consistent with these observations (Supplementary Fig. 11a). The interaction between RLP23 and BAK1 was detected upon treatment with the ligand nlp20 (*Pp*NLP), but this interaction was hardly detectable in mock-treated leaves. In the case of RXEG1, the association with SOBIR1 was detected regardless of XEG1 treatment (Fig. 6f, Supplementary Fig. 11b). However, we also repeatedly detected the interaction between RXEG1 and BAK1 even without XEG1 treatment (Fig. 6f). The interaction, however, was increased in XEG1-treated plants (Fig. 6f). These results demonstrate that some recruitment of BAK1 by RXEG1 occurs in a ligand-responsive manner and that RXEG1 forms a complex with BAK1 and SOBIR1, which may be involved in transducing the XEG1-triggered immune signal.

## Discussion

Membrane-localized receptors function on the frontline of plant defense. Elucidating how these receptors perceive microbial attack will significantly advance our understanding of plant innate immunity. LRR receptors are a common type of immune receptor in plants and animals[44]. In recent years, several PRRs were identified using map-based cloning or *Arabidopsis* T-DNA insertion lines[13,18,23–26]. Nevertheless, with the increasing number of identified MAMPs or elicitors able to induce plant immune responses[34,45], an efficient strategy to quickly pinpoint plant receptors is in high demand. *Arabidopsis* T-DNA insertion mutants have been used widely to identify PRRs or study the function of membrane-localized receptors[19,46,47]. However, the lack of homozygous T-DNA insertion lines as well as the off-target effects of T-DNA fragments often hampers gene function analysis. Moreover, the T-DNA insertion mutants are not suitable for analyses of genes that are functionally redundant or vital for plant growth and development. Last but not the least, some plant LRR receptor-like genes are lineage-specific and *Arabidopsis* lacks recognition receptors for some MAMPs, such as the LRR RLK CORE which recognizes csp22, the epitope of bacterial cold-shock proteins[29]. Therefore, we have developed a high-throughput toolkit for studying the LRR receptor-like genes using the model plant *N. benthamiana*, a close relative of tobacco that is amenable to VIGS. VIGS can be easily performed in *N. benthamiana* and enable efficient gene silencing within a month. Moreover, VIGS in *N. benthamiana* can avoid gene function redundancy and can achieve simultaneous silencing of multiple homologous genes with a sufficient efficiency. Therefore, VIGS in *N. benthamiana* overcomes the above-mentioned obstacles in *Arabidopsis*[48]. In this study, we generated 257 TRV-based constructs to silence all of the 403 identified genes that encode predicted membrane-localized LRR-RLPs and LRR-RLKs in

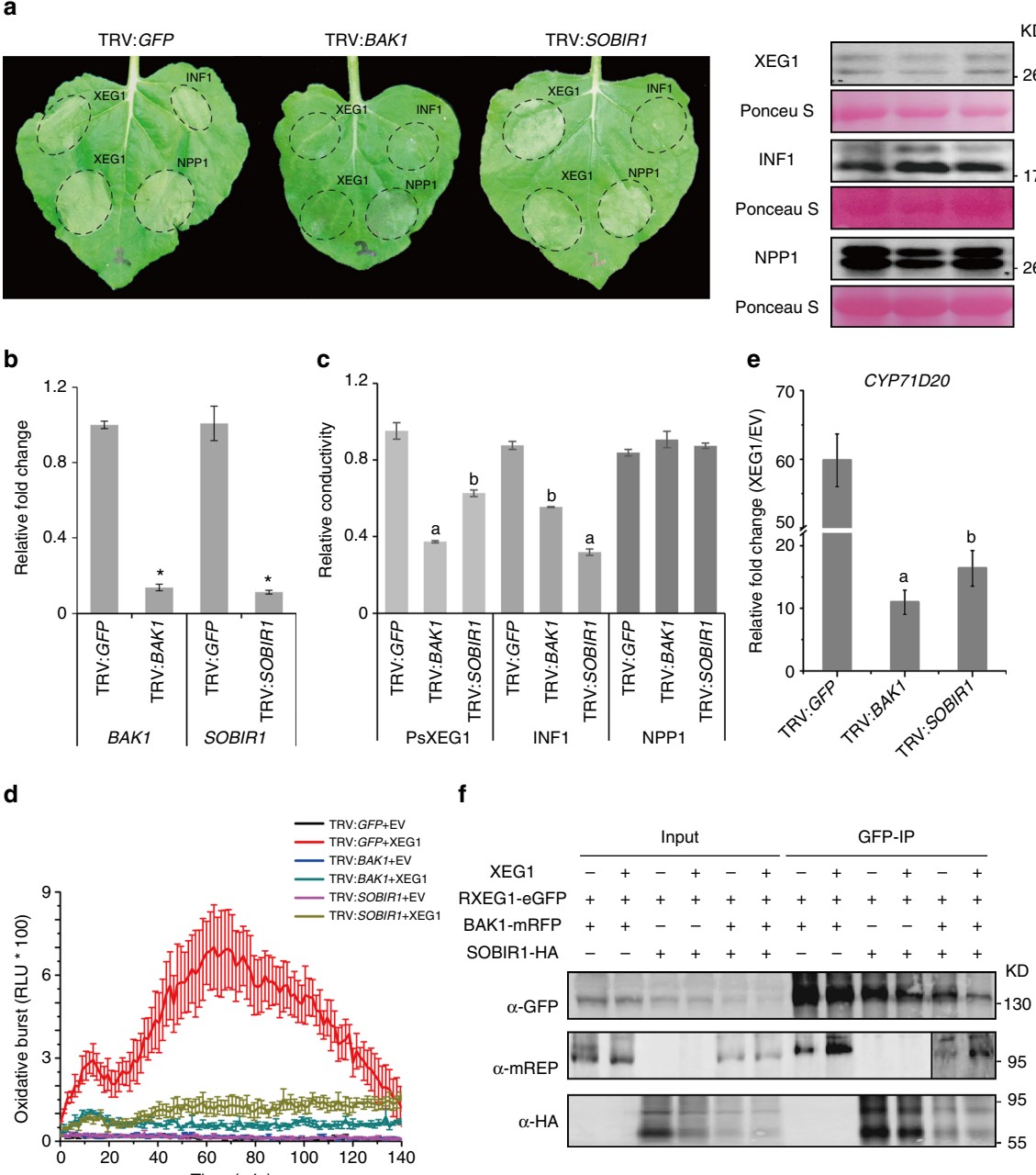

**Fig. 6** BAK1 and SOBIR1 associate with RXEG1 and are required for RXEG1-mediated XEG1 recognition. **a** Representative leaves showing cell death induced by expression of XEG1, INF1, or NPP1 in *N. benthamiana* leaves treated with TRV:*GFP*, TRV:*BAK1*, or TRV:*SOBIR1*. XEG1, INF1, and NPP1 were transiently expressed in *N. benthamiana*. Infiltrated leaves (*n* = 6) were photographed at 3 dai. Protein expression at 2 dai was monitored by western blotting with anti-HA antibody. Protein loading is indicated by Ponceau S staining. Experiments were repeated at least three times with similar results. **b** Silencing efficiency of *BAK1* and *SOBIR1* in *N. benthamiana*. The transcript levels were monitored by qRT-PCR, normalized with EF-1α, and expressed as mean fold changes (±s.e. m.) relative to TRV:*GFP*-treated leaves, which was set as 1. * indicates significant differences (*P* < 0.01, Student's *t*-test) in expression level when compared to that in TRV:*GFP*-treated plants. Experiments were repeated three times with similar results. **c** Ion leakage detected from *N. benthamiana* leaves expressing XEG1, INF1, or NPP1. Bars represent average value (±s.e.m.) of six replicates. In each treatment, different letters indicate significant differences (*P* < 0.01, one-way ANOVA). Experiments were repeated twice with similar results. **d** ROS production in *N. benthamiana* leaves treated with 1 μM XEG1 or EV. Mean values (±s.e.m.) of four replicates are shown. Experiments were repeated twice with similar results. **e** Relative expression of *CYP71D20* in *N. benthamiana* leaves treated with 200 nM XEG1 or EV. Transcript levels were quantified by qRT-PCR and normalized with EF1-α. Bars represent the mean fold changes (±s.e.m.) relative to the value in EV-treated leaves, which was set as 1. Different letters indicate significant differences (*P* < 0.05, one-way ANOVA). Experiments were repeated three times with similar results. **f** Interactions of RXEG1 with BAK1 and SOBIR1 in planta. *N. benthamiana* leaves were agro-infiltrated to express RXEG1-GFP, BAK1-RFP, or SOBIR1-HA and collected 2 dai after treatment with 1 μM XEG1 or EV for 10 min. Extracted proteins were subjected to Co-IP using GFP-trap® A beads and western blotting with anti-GFP, anti-RFP, and anti-HA antibodies

*N. benthamiana*. qRT-PCR analysis of the expression levels of 230 *LRR* receptor-like genes revealed that, in 199 of those genes, the silencing efficiency was greater than 60%. This collection of silencing constructs proved to be an effective tool for LRR receptor-like gene function analysis. On the one hand, silencing assays in *N. benthamiana* revealed that multiple previously unknown LRR-RLKs may be involved in plant growth; on the other hand, expression of the recently identified MAMP GH12 protein XEG1 in the silenced plants enabled the successful identification of the candidate recognition receptor RXEG1.

Our genetic and biochemical analyses demonstrated that RXEG1 is an essential component of the XEG1-recognition machinery. First, RXEG1 was shown to be essential for XEG1-induced cell death and immune responses and for similar responses to many other GH12 proteins. Second, overexpression of RXEG1 elevates plant responsiveness to XEG1. Third, the extracellular domain of RXEG1 which has 31 LRRs could associate with the ligand XEG1 in the apoplast as shown by Co-IP assays. In many other receptors, the extracellular LRR domains are specialized for ligand binding. A survey of previously well-documented LRR receptors revealed that PRRs often contain a high number of LRRs; for example, the flg22 receptor FLS2 has an ectodomain with 28 LRRs[49].

RXEG1, along with RXEGL1 and RXEGL2 are co-orthologs of the two tomato LRR-RLPs LeEix1 and LeEix2 (Supplementary Fig. 8b), which involved in recognition and response to the fungal elicitor EIX[21]. Nevertheless, LeEix1 is not required for EIX recognition. Instead, LeEix1 heterodimerizes with LeEix2 to downregulate LeEix2 signaling following EIX recognition in *Nicotiana tabacum*[50]. However, our Co-IP assays demonstrated that RXEG1 associates specifically with XEG1 and does not associates with fungal EIX in planta. Consistent with this, fungal EIX failed to trigger cell death and immunity in *N. benthamiana*, demonstrating that *N. benthamiana* lacks a functional EIX recognition receptor. Moreover, silencing of *LeEix2* in tomato did not affect XEG1-induced plant cell death in tomato[34]. Taken together, our results demonstrate that RXEG1 and LeEix2 are co-orthologous receptors that have evolved to perceive related but different MAMPs.

RXEG1 belongs to the LRR-RLP-type of receptors whose function is usually highly dependent on the common adaptor SOBIR1[30]. One exception is *N. benthamiana* CSPR, which constitutively associates with SOBIR1 but does not require SOBIR1 for its function in csp22 responses[28]. In the case of RXEG1, however, cell death induced by XEG1 was only slightly compromised when *SOBIR1* was silenced, indicating that the ability of RXEG1 to trigger cell death is not solely dependent on SOBIR1. Nevertheless, we found that RXEG1 constitutively associates with SOBIR1 and requires SOBIR1 for immune signal transduction such as ROS production and *CYP71D20* induction. In addition to SOBIR1, the LRR-RLK BAK1 is also required for RXEG1-mediated XEG1 response. BAK1 associates with multiple LRR-RLK-type PRRs including FLS2 and CORE1 and those interactions only occur in the presence of the cognate ligand[29,51]. The interaction between BAK1 and RLPs is rather variable. RLP23, which recognizes nlp20 interacts with BAK1 in an MAMP-dependent manner and requires BAK1 for signaling [24]. In contrast, tomato LeEix2 failed to interact with BAK1 in planta regardless of ligand treatment[50]. Similarly, *Arabidopsis* RLP42, which recognizes fungal endopolygalacturonases, also did not associate with BAK1 in planta and functions independently of BAK1[25]. These receptors might associate with other RLKs, such as different SERKs, to transduce recognition signal. In this study, we found that RXEG1 interacts with BAK1 in planta even without XEG1 elicitation. This is consistent with a previous report that a

co-ortholog of RXEG1 in tomato (i.e., LeEix1) associates with BAK1, which was confirmed by yeast two-hybrid and bimolecular fluorescence complementation assays[50]. Nevertheless, the interaction between RXEG1 and BAK1 was increased upon XEG1 treatment, demonstrating that XEG1 can significantly promote the RXEG1−BAK1 interaction. FLS2 is another PRR that forms a complex with BAK1 upon ligand (i.e., flg22) treatment. According to the solved structure, the FLS2LRR-flg22-BAK1LRR complex shares two interfaces[14]. BAK1LRR binds directly to FLS2LRR, which may account for the flg22-independent FLS2−BAK1 association detected in some cases[52]. The other interface is the ligand flg22-dependent bridging of FLS2LRR and BAK1LRR, which promotes the heterodimerization of FLS2LRR and BAK1LRR. Whether the RXEG1-XEG1-BAK1 complex forms in a similar manner, and the detailed molecular mechanisms thereof, remain to be uncovered.

Based on current knowledge, BAK1 and SOBIR1 participate solely in signaling mediated by LRR receptors, but not receptors with other types of ectodomains such as LysM-receptors or lectin receptor kinases[53,54]. Recently, multiple novel MAMPs were identified that require BAK1 or both BAK1 and SOBIR1[45,55], indicating that the corresponding receptors may belong to the LRR class. Moreover, the recognition of certain MAMPs is restricted in Solanaceous plants[29,55]. Therefore, the silencing of the LRR receptor candidates should be a straightforward approach to identify recognition receptors for each MAMP, which will ultimately lead to improved plant resistance. In addition, as LRR receptors belong to a conserved receptor family among plant species, functional analysis of these receptors in *N. benthamiana* will also improve our understanding of their function in other plant species, especially in those species in which genetic manipulation is difficult.

## Methods

**Sequence identification and analyses**. An HMM search by the software HMMER (version 3.0; with default parameter) using LRR (PFAM ID: PF00560.31; downloaded from http://pfam.xfam.org) was performed via the Sol Genomic Network (SGN) website (http://solgenomics.net) against the genomic databases of *N. benthamiana* (V1.0.1) or *Solanum lycopersicum* (ITAG release 3.20). All non-redundant protein sequences were retrieved and analyzed with Conserved Domains (https://www.ncbi.nlm.nih.gov/Structure/cdd/wrpsb.cgi), SMART (http://smart.embl-heidelberg.de), Pfam (http://pfam.xfam.org/), and SignalP 3.0 (http://www.cbs.dtu.dk/services/SignalP-3.0) for protein domain and motif prediction. Proteins with both LRR and TM domains were considered as potential LRR receptors. For *N. benthamiana*, both protein and cDNA sequences were retrieved and the obtained cDNA sequences were further validated using RNA-sequencing data derived from the QUT *Nicotiana benthamiana* Genome & Transcriptom Sequencing Consortium (http://benthgenome.qut.edu.au/)[56,57].

**Phylogenetic analysis**. For LRR-RLKs, phylogenetic maximum-likelihood (ML) tree was constructed using the predicted kinase domain sequences and PhyML implemented in SEAVIEW software (http://doua.prabi.fr/software/seaview). The *N. benthamiana* LRR-RLK candidates were classified into different subfamilies according to the reference LRR-RLKs in *Arabidopsis thaliana*, which represents the subfamilies I–XV[5,6]. For LRR-RLPs, phylogenetic tree was constructed using the conserved C3 and D domains[58,59]. The phylogenetic trees were displayed using iTOL (http://itol.embl.de).

**Plasmid construction**. Fragments used to generate silencing or overexpression constructs were amplified by PCR from cDNA or genomic DNA of *N. benthamiana*, *Phytophthora* pathogens, or *Verticillium dahliae* using PrimeSTAR GXL DNA Polymerase (Takara Bio Inc., Otsu, Japan) with the primers listed in Supplementary Table 1. The purified fragments were cloned into the modified gene-silencing vector pTRV-RNA2[60] or the expression vectors pBIN-(c)eGFP, pBIN-(c)HA, pBIN-(c)mRFP using the ClonExpress II One Step Cloning Kit (Vazyme Biotech Co., Ltd, China). The full length coding sequences containing the SP were amplified for INF1, NPP1, and XEG1 homologs except Avh241 without SP[61]. The full length coding sequence[62] of EIX was synthesized in Genscript. All these fragments were subsequently cloned into vector pGR107-cHA. The resulting vectors were verified by sequencing and individually transformed into the *Agrobacterium tumefaciens* strain GV3101. TRV:BAK1[51] and TRV:SOBIR1[63] were shown in supplementary Data 1.

**Plant growth conditions and infection assays**. *Nicotiana benthamiana* plants were grown in soil in a conditioned climate chamber at 19–22 °C with a 14 h photoperiod and 70–80% relative humidity. *P. parasitica* 025 was maintained in the dark at 25 °C on 20% (v/v) V8 juice agar[64]. Leaf infection assays were performed as follow[65]: leaves were collected from *N. benthamiana* with similar plant sizes at the same position. Inoculated leaves were kept in transparent plastic boxes with high humidity and placed in the climate chamber. Inoculated leaves were kept in the dark for the first 24 h. Lesion diameters were measured at 3 days post-inoculation (dpi).

**RNA isolation and quantitative reverse transcription PCR**. Leaves were selected from the same position of *N. benthamiana* of similar size and ground in liquid nitrogen. Total RNA was isolated from 100 mg of ground material using an RNA Kit (Omega Bio-tek, USA) and used as a template for first strand cDNA synthesis using PrimeScript Reverse Transcriptase (Takara Bio Inc.). Quantitative reverse transcription PCR (qRT-PCR) was performed on an ABI 7500 Fast Real-Time PCR system (Applied Biosystems Inc., Foster City, CA, USA) using PrimeScript RT Master Mix (Takara Bio Inc.) with translation elongation factor 1 alpha (EF-1α) as an endogenous control and the primers listed in Supplementary Data 1 or Supplementary Table 1. Data were analyzed using the $2^{-\Delta\Delta CT}$ method[66].

**Quantification of *Phytophthora* biomass**. *Nicotiana benthamiana* leaf discs ($d = 4.0$ cm) were collected from infection sites 3 days after *P. parasitica* inoculation. The leaf discs were ground in liquid nitrogen and used for DNA isolation using the genomic DNA extraction kit (TIANGEN Biotech, Beijing, China). The eluted products were used as templates for qPCR with *N. benthamiana* EF-1α as an endogenous control, while the primers PAR-F/R were used to target *P. parasitica* (Supplementary Table 1). qPCR and data analyses were performed as described above.

**Agro-infiltration assays in *N. benthamiana***. *Agrobacterium tumefaciens* strains carrying various vectors were grown overnight in LB medium with the appropriate antibiotics. *Agrobacterium tumefaciens* cells were pelleted, resuspended, and incubated in infiltration medium (10 mM 2-[N-morpholino] ethanesulfonic acid pH 5.6, 10 mM MgCl₂, 200 µM acetosyringone) for 3–4 h. For the TRV-mediated silencing assay, *A. tumefaciens* cultures expressing pTRV-RNA2 constructs were mixed in a 1:1 ratio with *A. tumefaciens* culture expressing pTRV1 to a final $OD_{600}$ of 1.0 before infiltration into cotyledons of four-leaf-stage *N. benthamiana* seedlings. TRV:PDS and TRV:GFP were used as controls. For transient gene expression, *A. tumefaciens* suspensions were syringe-infiltrated into fully expanded leaves of six-week-old *N. benthamiana* with appropriate concentrations ($OD_{600} = 0.2$ for NPP1, INF1, Avh241, EIX, XEG1, Ps138787 and PITG06962; $OD_{600} = 0.6$ for Ps119627, PPTG_16272, and VDAG_07406; $OD_{600} = 1.0$ for RXEG1, RXEGLs, RLP23 and derivatives). For co-infiltration with the silencing suppressor P19, suspensions were mixed in an appropriate ratio to a final $OD_{600}$ of 0.6.

**Co-immunoprecipitation and western blotting**. Agro-infiltrated leaves were collected and ground in liquid nitrogen. Total protein was extracted by incubating the ground leaf samples in extraction buffer containing 150–200 mM NaCl, 50 mM Tris-HCl pH 7.5,10 mM ethylenediaminetetraacetic acid, 1.0% (v/v) NP-40, 1 mM phenylmethylsulfonyl fluoride, and 0.1% (v/v) protease inhibitor cocktail (P9599; Sigma, St. Louis, MI, USA) for 30 min. The homogenate was centrifuged at 21,000×g for 15 min and the supernatant was incubated with GFP-trap® A beads (Chromotek, Hauppauge, NY, USA, gta-20) at 4 °C for 2 h. The beads were pelleted and washed with extraction buffer for six times. The proteins were then eluted from the beads by boiling for 10 min, separated by 10% sodium dodecyl sulfate polyacrylamide gel electrophoresis, and transferred to a polyvinylidene fluoride (PVDF) membrane (Bio-Rad, Hercules, CA, USA). The PVDF membrane was blocked with phosphate-buffered saline (PBS) pH 7.4 with 5% skim milk for 1 h at room temperature followed by three washes with PBS with 0.1% Tween-20. Co-immunoprecipitated proteins were analyzed by incubating the membrane with 1:5000 diluted anti-GFP (Abmart, Berkeley Heights, NJ, USA, M20004L), anti-HA (Abmart, Berkeley Heights, NJ, USA, M20003L), or anti-RFP (Abcam, Cambridge, UK, ab62341) antibodies followed by incubation with 1:10,000 diluted goat anti-mouse (irdye 800, 926-32210; LI-COR Biosciences, Lincoln, NE, USA) or goat anti-rabbit (irdye 800, 926-32211; LI-COR Biosciences, Lincoln, NE, USA) antibody. The blots (Supplementary Fig. 12) were visualized using the Odyssey® LI-COR Imaging System. Equal protein loading was confirmed with Ponceau S stain.

**Expression and purification of recombinant XEG1 protein**. pPICZα vector containing XEG1-His[34] or empty vector was transformed into *Pichia pastoris* strains X33 (Muts⁺) or KM71 (Muts). *P. pastoris* was cultured overnight in the YPD medium at 30 °C and subsequently grown in the BMGY (Buffered Glycerol-Complex Medium) and BMMY (Buffered Methanol-Complex Medium) (pH = 6.5) for protein expression. The recombinant protein XEG1-His or EV was purified from the supernatant of the *P. pastoris* culture harboring pPICZα-XEG1-His or

pPICZα using the AKTA™ avant 25 (GE Healthcare) through the HisTrap™ FF Ni Sepharose Columns (5 ML, 17525501; GE Healthcare) and HiTrap™ Desalting Prepacked Columns (5 ML, 29048684; GE Healthcare).

**Measurement of reactive oxygen species**. Reactive oxygen species (ROS) production was monitored with a luminol/peroxidase-based assay on leaf discs (Ø 0.5 cm) collected from 5-week-old *N. benthamiana* plants and floated overnight in 200 µL of sterile H₂O in a 96-well plate. H₂O was replaced with the luminol (35.4 µg/mL)/peroxidase (10 µg/mL) reaction solution supplied with sterile water, 100 nM flg22 (Genscript Biotech Corporation, China), 1 µM purified XEG1 protein, or EV. Luminescence was measured using the GLOMAX96 microplate luminometer (Promega, Madison, WI, USA).

**Electrolyte leakage assay**. Six *N. benthamiana* leaf discs (Ø 1.0 cm) were collected 3 days after agro-infiltration and floated on 4 mL of deionized water for 3 h with continuous shaking (100 rpm) at room temperature. The initial electrolyte leakage values and the final electrolyte leakage values after 15 min of boiling were measured using a conductivity meter (Con 700; Consort, Tutnhout, Belgium). Relative electrolyte leakage was calculated by comparing the initial and final electrolyte leakage values.

**Data availability**. Sequence information can be found in GenBank with the following accession numbers: RXEG1 (MG010652), RXEGL1 (MG010653), and RXEGL2 (MG010654). All data relevant to this study are available from the corresponding author upon reasonable request.

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

## Acknowledgements

We thank J-M. Zhou and G-X. Wang (Chinese Academy of Science), W. Ma (University of California-Riverside) for helpful suggestions; we also thank Y. Wang, B. Guo, Y. Xia, Y. Shi and H. Zhang (Nanjing Agricultural University) and Q. Wang (Northwest A&F University) for technical assistance. This work was supported in part by grants to Y. Wang from the China National Funds for Innovative Research Groups (Grant No.31721004), the Special Fund for Agro-scientific Research in the Public Interest (201303018), and the China Agriculture Research System (CARS-004-PS14); by a 111 International Cooperation grant (B07030) to Nanjing Agricultural University from the Chinese government; and by grants to Y. Wang from the National Natural Science Foundation of China (31501622) and from Nanjing Agricultural University (KJQN201663).

## Author contributions

Y.W., S.D., and Y.C.W conceived the idea and designed the experiments; Y.W., Y.X., Y.S., H.W. J.Q., Y.S., B.W., Y.L. and W.Y. performed the experiments; Y.W., Y.X., Y.S., H.W, J. Q., and W.Y., S.D., B.M.T. and Y.C.W analyzed and interpreted the data; Y.W., S.D., B.

M.T. and Y.C.W. wrote the manuscript. All authors read and approved the final manuscript for publication.

## Additional information

**Competing interests:** The authors declare no competing financial interests.

