## [Peer Review File · Nature Communications]

Reviewers' comments:

Reviewer #1 (Remarks to the Author):

XEG1, the glycoside hydrolase (GH) 12 protein, was previously identified as a PAMP. Overexpression of XEG1 or exogenous infiltration of purified XEG1 in tobacco triggered cell death. To identify the putative plant receptors (or components) required from XEG1 perception, in this ms, the authors have used a VIGS approach to transiently silence individual LRR domain containing receptor-like proteins (LRR-RLPs) or receptor-like kinases (LRR-RLKs) and screened for blocking the cell death-inducing activity triggered by XEG1. Among the candidate genes they screened, RXEG1 (Receptor for XEG1), an LRR-RLP is required for XEG1-, but not the other PAMPs, triggered cell death, ROS production and one marker gene expression. They further showed that RXEG1 co-immunoprecipitated with XEG1, likely via the LRR motif. In addition, XEG1-triggered responses require RXEG1, RXEG1 associates with BAK1 and SOBIR1 and XEG1 promotes RXEG1 and BAK1 association. It appears that XEG1 perception shares a similar theme as the other plant LRR-RLP-mediated signaling. The work exemplifies a unique approach to identify the receptors and signaling components for PAMPs and the genome-wide collection of VIGS-RLPs and RLKs in tobacco is useful to explore additional PAMP receptor functions. The work was logically designed and the ms was clearly written except a few places (see below).

Based on the data that silencing of RXEG1 blocked XEG1-triggered cell death, the authors stated that (Page 6. Line 203) RXEG1 is the genuine receptor responsible for XEG1-recognition. To now, the authors could only reach a conclusion that RXEG1 is genetically required for XEG1-triggered responses. In the entire ms, the data to support RXEG1 and XEG “interaction” are based on in plants co-IP assays. No direct binding experiments are shown. It’s necessary that the authors modify the presentation and avoid any potential overstatement (line 288: in this subtitle and following content, the author claimed that RXEG1 binds to GH12 protein; the co-IP data in fig4 and 5 only provided the evidence of association).

1. Typo in the title: Receptormics should be Receptoromics. I am conserved to use this term as they may not real receptors rather as signaling components. receptor-like proteins are scientifically more precise.
2. PAMPs function outside of the plant cells and are perceived by the extracellular motif of the plasma membrane-resident receptors. Usually, PAMP treatment was done by infiltration of purified PAMPs into the intercellular space. I don’t understand how come expression of various PAMPs, including XEG1, INF1, NPP1, Avh241, or EIX, inside of plant cells could trigger

responses (fig4b). did the authors add secretion signals in these constructs? It has been shown that expression of the well-known PAMP flagellin gene in plant cells does not trigger responses as seen by infiltration of flagellin (Alan Collmer's group).

3. The presentation on the genome-wide analysis of LRR-RLKs and LRR-RLPs (result section 1) was not clearly presented. Figure 1 is not so informative. The authors could consider to include a table to list the numbers of LRR-RLPs and LRR-RLKs in different plant species. Figure 1a. Legend is not clear. What are RD, CD, GD...2. Fig1b is not easy to read. the colors for several categories are hard to differentiate. The labeling and dotted line are merged together and not indicative. It's better to re-organize this fig. 1c could be in supplemental fig.

4. There appears a tendency that when RXEG1 is reduced, RXEGL1, L2 are higher (Fig2b). RXEGL1 expression in TRV:RXEG1-3 is ~2 fold of that in TRV:GFP. Not sure whether it will pass the statistic, but this data confused me that silencing RXEG1 would induce RXEGL1 expression.

5. The molecular weight of INF1 in Fig 2a and d are different.

6. Fig 3. Since RXEG1 and RLP23 showed enhanced R to Phytophthora infection (fig 3e), why there are no elevated ROS in fig 3c in these plants (RXEG1-EV)? In WT plants, there is the endogenous RXEG, why there were no enhanced ROS in WT upon treatment of XEG (WT-XEG) in fig 3c?

7. Fig4a: In both anti-GFP Input and GFP-IP panels, the RXEGL2-eGFP expression is hard to see, so It's unable to conclude the interaction between XEG1 and RXEGL2 present or not.

8. Fig5a: The RXEG1 (SP+LRR) truncated proteins should be secreted and lost the membrane localization anymore, how will it transduce the signal into cytosol? It may be more informative to elucidate which LRRs are responsible for the association by mutations instead of these truncations.

9. Fig6 and Supplementary Fig5: BAK1 contributes to the recognition of XEG1. How about its disease susceptibility to Phytophthora?

10. Line101: 1A should be in lower-case.

11. Line 135: It's better to list how many RLPs and RLKs are screened in these 300 LRR genes. Supplementary Fig2a: there is no labeling for RXEG1(S)-eGFP or eGFP expression in the tobacco leaves.

12. No data to show that SOBIR1 is a co-receptor. Modify writing.

Reviewer #2: (Remarks to the Author):

Review of Wang et al., Nature Communications 2017

The authors present a genome-wide analysis of genes encoding leucine-rich repeat proteins (RLPs) and receptor-like kinases (RLKs) in the model plant species *Nicotiana benthamiana*. The

bioinformatics appears to be comprehensive and convincing and resulted in the development of 274 TRV constructs used for gene silencing of all 415 relevant genes. One of the targeted genes, RXEG, is shown to play a role in recognition of a oomycete MAMP (XEG) and, importantly, complementation analysis with a synthetic version of the gene in a RXEG-silenced plant confirmed the gene's involvement in several host responses to XEG. The authors over interpret their data here by saying they have identified the 'receptor' for XEG (on lines 181, line 196 and line 203). In fact they can only conclude at this point that RXEG is required for the host cell death response to the MAMP. They later tone down these statements by saying that RXEG 'potentiates the response to XEG' or 'contributes to plant immunity' (lines 257 and 269).

The evidence for RXEG acting as a receptor for XEG comes in the subsequent section where assays showed that the RLP co-IPs with XEG with some specificity. It does not co-IP with several other MAMPs, including EIX which is bound by tomato LeEIX2 which encodes the tomato ortholog to RXEG. Finally, although silencing of SOBIR only mildly affects the host response to XEG RXEG does associate with the co-receptor SOBIR in a ligand-independent manner. Overall, the experiments are well-conceived, technically good with appropriate controls, and the results are straightforward and convincing.

The paper describes an enormous effort involving bioinformatics, cloning and testing of a large number of TRV constructs, and subsequent assays of the plants with XEG. The results are not especially novel since all of the methods are widely used and there is nothing especially unusual about RXEG (i.e., several other RLPs are known to play a role in plant immunity). However, if the clones are easily available from the authors (this needs to be made clearer) then they constitute a valuable community resource that could greatly impact the PRR field and this paper would be widely cited.

A few minor comments:

Title: Is there a typo here? Should it be receptoromics? Regardless, I would suggest something more descriptive like: Genome-wide analysis of leucine-rich repeat-encoding genes identifies RXEG as the receptor for the fungal glycoside hydrolase 12 protein.

Lines 49-64: A citation to the recently-identified FLS3 gene would be relevant here. Hind, S.R., et al., (2016). *Nature Plants* 2:16128.

Line 77: Change to Solanaceous plant family.

Line 77: Provide a citation to the high genomic collinearity of these three species (I am not aware of one).

Line 129: A citation for the VIGS Tool should be provided. Fernandez-Pozo, N., et al., (2015). The SGN VIGS Tool: User-friendly software to design virus-induced gene silencing (VIGS) constructs for functional genomics. *Molecular Plant* 8:486-488.

Line 134: I did not see an explicit statement about how the 274 TRV clones will be made available to the community and that is essential for the paper to have serious impact.

Line 137: Some more descriptive language is needed here to define “silencing efficiency”. Presumably they looked at transcript abundance, compared it to a *N. benthamiana* infected with TRV-GFP only and then normalized the data to a control gene known to be stably expressed?

Lines 181, 196, 203: Modify all these statements about RXEG being a receptor since that is not supported by the data at this point in the manuscript.

Line 457: The relationship between LeEIX1, LeEIX2 and RXEG needs to be further clarified. Are the two former genes the most closely related to RXEG in the tomato genome? Are they likely orthologs? In this regard I think it would be useful for the authors to provide an amino acid alignment of these three proteins in the supplemental data section.

Line 462: receptors should be co-receptors.

Reviewer #3 (Remarks to the Author):

In this manuscript, the authors describe a toolkit aiming at silencing genes encoding LRR-containing receptors in *Nicotiana benthamiana*. While no insertion libraries exist to disrupt *N. benthamiana* genes and while only few genetic resources exist for this plant species, establishing a collection of silencing fragments targeting a particular gene family is of particular interest. The author then describe the use of this tool to identify a LRR-receptor-like protein (LRR-RLP) for responses triggered by the *Phytophthora sojae* XEG1 elicitor, which they called RXEG1. They also demonstrate that RXEG1 is able to perceive paralogous elicitors from other *Phytophthora* species and from the fungus *Verticilium dahliae*.

Here are some specific comments on the manuscript, which should help improving this manuscript:

- The authors present an interesting toolkit and also convincing data about the characterisation of RXEG1. However, a detailed phylogenetic analysis is missing as the authors describe for the first

time the *Nicotiana benthamiana* LRR-RLKs and LRR-RLPs. The members of the LRR-RLKs family are usually classified in sub-families using the *Arabidopsis* proteins as reference (as in the recent analysis published by Dufayard et al., 2017 doi:10.3389/fpls.2017.00381, for example). However, in the present manuscript, the authors simply define OrthoGroups (based on unknown parameters) and describe “highly homologous” sequences rather than orthologs. Also, because no phylogeny has been constructed, no candidate orthologs of the *N. benthamiana* RXEG1 can be presented nor discussed in this manuscript.

Furthermore, indicating that “RXEG1 is highly homologous to LeEIX1” is misleading and would suggest that LeEIX is the closest ortholog. Also, it would be preferable to perform a classical phylogenetic analysis from which the readers could benefit.

- The Table S1 is of major importance in this manuscript, thus a particular effort should be given to make it faultless. For example, the column “Full length aa” refers to unknown indications such as LRR, correct or corrected; the number of LRR is sometime missing; one target is of type “TM”; decimals are presented with 9 digits instead a 2 or 3; NbS00001134g0002 appears as sequence under T2, but appears “corrected” under T40; “undet” is missing for plenty of silencing efficiency; NbS00003987g0011, NbS00004687g0019 and NbS00037942g0004 have a “RT primer-R” but no “RT primer-F”; SP are alternatively indicated are lacking, no, no clear; types described as “LRR” would be secreted?

- While the data presented clearly show that RXEG1 is required for XEG1 responsiveness, it is not possible for the author to state that it is the receptor, unless data showing direct binding of XEG1 are shown (it would be in this context also useful to calculate binding affinity values). Co-immunoprecipitation is not a proof of direct binding. As such, lines 203-204 should, for example, currently rather state “...RXEG1 is required for XEG1-triggered cell death in *N. benthamiana*.”

- Title: the current title does not provide much context. I would suggest “Leucine-rich repeat receptoromics analysis reveals the *Nicotiana benthamiana* RXEG1 detects the PAMP glycoside hydrolase 12 protein”.

- Abstract: line 27: should be “...a complex with the LRR-receptor kinases BAK1 and SOBIR1...”.

- It is not entirely correct to refer to SOBIR1 as a co-receptor as it does not directly bind the ligand. Rather, it is proposed that LRR-RLPs form constitutive complexes with SOBIR1 to form what would constitute a bi-partite LRR-RLK, which could then form a ligand-induced complex with the co-receptor BAK1. This should make clearer in the Introduction and the rest of the manuscript. Also, the authors should clearly state that SOBIR1 only plays a role for the function of LRR-RLPs, which is not necessarily clear currently from the Introduction (eg. line 60).

- It is not necessarily appropriate to name NbS00035240g0005 as RXEGL2 given that it has only 12 LRRs compared to 26 for RXEG1 and RXEGL1.
- Decide whether you'd like to use the MAMP or PAMP abbreviation. At the moment, both abbreviations are used throughout the manuscript.
- Fig 1c: For a better reading, (and using a clock analogy) the values should start with 50 at 12:00 rather than at 10:00.
- Fig 3a, 3c, 6d, S4a: in the panels, please replace "TRV:GFP-EV" with "TRV:GFP + EV", etc ...
- Fig S1b: Percentages should be followed by the ID of the corresponding genes. Also the corresponding legend is missing for this figure panel.
- Fig S6: In the fifth line, RXEGL1 should be indicated rather than RXEG1. It is unclear what the differences are between SP+LRR and SP+LRR(s). For RXEGL1SP+TM-tail, 910 may be missing to indicate that the blue part is the eJM.
- Please indicate which HMM models have been used for LRR and TM.
- Please list and describe all the elicitors used in the manuscript (nlp20, EIX, Ps119627, Ps138787, Avh241, ...) in the Methods section.
- It is unclear how 1 μ M XEG1 is obtained. Is GenScript supplying a peptide, protein or gene? Is the protein tagged?
- line 36-37 : since all MAMPs are "elicitors released from microbes", the sentence should be rephrase.
- line 39: "plethora" should be replace with a more accurate range of gene numbers.
- line 43: Please rephrase "divergent in their ectodomain".
- line 65: should be "sequences microbial genomes".
- line 80: Please replace "Multiple PAMPs and elicitors" with "Multiple PAMPs including elicitors".
- line 99-101: The sentence need to be rephrased. It is unclear why 4 LRR-Ser/Thr kinase needed

corrections.

- Line 103: no results are presented to support the existence of “two copies of highly homologous LRR receptors”.
- line 135: Please replace “Table 1” with “Table S1”.
- line 148: should be “Phylogenetic analysis”.
- line 175 and 193: should be “infiltrated” rather than “treated”.
- line 182: Please rephrase as the target genes could also encode regulatory LRR-RLK/P.
- line 197: The two sentences should be merged.
- line 203: Please rephrase the sentence as RXEG1 requirement doesn't demonstrate its role as bona fide receptor.
- lines 244-245: rewrite to “...and confers disease resistance”.
- line 255-257: should be “over-expression”.
- line 269: should be “to confer resistance to Phytophthora when expressed in *N. benthamiana*”.
- line 297: As RXEGL2-eGFP is not detected after GFP-IP in Fig 4a. Also, it can't be stated that this receptor doesn't interact with XEG1-HA.
- line 301: Please rephrase, as it is confusing to present the tomato homologs of RXEG1 as no phylogenetic analyses have been performed.
- line 307: There is no dependent relation between EIX not interacting with RXEG1 and EIX not being perceived in *N. benthamiana*. Please rephrase.
- line 329: It may be indicated that SP-TM-tail is in fact SP-eJM-TM-tail.
- lines 356 and 486: see comment above about SOBIR1 and the term co-receptor.
- line 368: should be “associated with LRR-type PRRs”.
- line 432: Please replace “LRR receptor” with “LRR transmembrane receptor”.

- line 458: Please replace “receptors function” with “receptors whose function”.
- lines 458-463: it should be note here that another LRR-RLP, CSPR, was also previously shown to associate with SOBIR1 but not depend on it for its function (Saur et al., 2016).
- line 467: “variable” may be used instead of “controversial”. Also, it may be worthwhile here to mention that these results might suggest that other SERK proteins are preferentially used by these receptors.
- line 497: Please replace “for those are” with “for those which are”.
- line 507: Please indicate which versions of the genome releases were used for sequence analysis.
- line 529: References for expressions vectors are missing.
- line 531: The reference to EIX sequence is presented as Rotblat et al., 2002. However, the TvEIX sequence was rather described in Furman-Matarasso et al., 1999.

Response to Reviewers' comments:

Reviewer #1 (Remarks to the Author):

XEG1, the glycoside hydrolase (GH) 12 protein, was previously identified as a PAMP. Overexpression of XEG1 or exogenous infiltration of purified XEG1 in tobacco triggered cell death. To identify the putative plant receptors (or components) required from XEG1 perception, in this ms, the authors have used a VIGS approach to transiently silence individual LRR domain containing receptor-like proteins (LRR-RLPs) or receptor-like kinases (LRR-RLKs) and screened for blocking the cell death-inducing activity triggered by XEG1. Among the candidate genes they screened, RXEG1 (Receptor for XEG1), an LRR-RLP is required for XEG1-, but not the other PAMPs, triggered cell death, ROS production and one marker gene expression. They further showed that RXEG1 co-immunoprecipitated with XEG1, likely via the LRR motif. In addition, XEG1-triggered responses require RXEG1, RXEG1 associates with BAK1 and SOBIR1 and XEG1 promotes RXEG1 and BAK1 association. It appears that XEG1 perception shares a similar theme as the other plant LRR-RLP-mediated signaling. The work exemplifies a unique approach to identify the receptors and signaling components for PAMPs and the genome-wide collection of VIGS-RLPs and RLKs in tobacco is useful to explore additional PAMP receptor functions. The work was logically designed and the ms was clearly written except a few places (see below).

Based on the data that silencing of RXEG1 blocked XEG1-triggered cell death, the authors stated that (Page 6. Line 203) RXEG1 is the genuine receptor responsible for XEG1-recognition. To now, the authors could only reach a conclusion that RXEG1 is genetically required for XEG1-triggered responses. In the entire ms, the data to support RXEG1 and XEG "interaction" are based on in plants co-IP assays. No direct binding experiments are shown. It's necessary that the authors modify the presentation and avoid any potential overstatement (line 288: in this sub-title and following content, the author claimed that RXEG1 binds to GH12 protein; the co-IP data in fig4 and 5 only provided the evidence of association).

Response: We agree with the reviewer and toned down our statement on RXEG1. Based on the experimental data, we concluded that RXEG1 is essential for XEG1-recognition and changed 'binds' to 'associates with' throughout the manuscript.

1. Typo in the title: Receptormics should be Receptoromics. I am conserved to use this term as they may not real receptors rather as signaling components. receptor-like proteins are scientifically more precise.

Response: We have changed the title to 'Leucine-rich repeat receptoromics screen reveals that RXEG1 detects glycoside hydrolase 12 MAMPs in *Nicotiana benthamiana*'. Furthermore, throughout the revised paper, we have also been careful to distinguish receptor-like family proteins from receptors with proven ligand binding and signaling properties.

2. PAMPs function outside of the plant cells and are perceived by the extracellular motif of the plasma membrane-resident receptors. Usually, PAMP treatment was done by infiltration of purified PAMPs into the intercellular space. I don't understand how come expression of various PAMPs, including XEG1, INF1, NPP1, Avh241, or EIX, inside of plant cells could trigger responses (fig4b). did the authors add secretion signals in these constructs? It has been shown that expression of the well-known PAMP flagellin gene in plant cells does not trigger responses as seen by infiltration of flagellin (Alan Collmer's group).

Response: We were sorry that we did not make this clear in the previous manuscript. In this study, the full length coding regions of XEG1, INF1, NPP1, or EIX, including their native signal peptides were cloned into the pGR107 vector for expression in *N. benthamiana*. We added this description in the Methods section in this revised manuscript.

3. The presentation on the genome-wide analysis of LRR-RLKs and LRR-RLPs (result section 1) was not clearly presented. Figure 1 is not so informative. The authors could consider to include a table to list the numbers of LRR-RLPs and LRR-RLKs in different plant species. Figure 1a. Legend is not clear. What are RD, CD, GD...2. Fig1b is not easy to read. the colors for several categories are hard to differentiate. The labeling and dotted line are merged together and not indicative. It's better to re-organize this fig. 1c could be in supplemental fig.

Response: The numbers of LRR-RLPs and LRR-RLKs in *N. benthamiana* are listed in Table 1. LRR-RLKs are also classified into different subgroups according to the description by Shiu et al (Plant Cell, 2004). These data are now shown in Table 1, Supplementary Table 2, and supplementary Figure 1. We removed the analyses of LRR receptors from other plant species so that this part of the message fits better with this manuscript.

4. There appears a tendency that when RXEG1 is reduced, RXEGL1, L2 are higher (Fig2b). RXEGL1 expression in TRV:RXEG1-3 is ~2 fold of that in TRV:GFP. Not sure whether it will pass the statistic, but this data confused me that silencing RXEG1 would induce RXEGL1 expression.

Response: We tested different *N. benthamiana* plants treated with TRV:RXEG1-3 in the other two biological replicates (as shown below) and we could not find clear correlation in expression levels between RXEG1 and RXEGL1.

5. The molecular weight of INF1 in Fig 2a and d are different.

Response: We are sorry to make this confusion. We changed the figure and label accordingly to make it consistent in the two figures.

6. Fig 3. Since RXEG1 and RLP23 showed enhanced R to Phytophthora infection (fig 3e), why there are no elevated ROS in fig 3c in these plants (RXEG1-EV)? In WT plants, there is the endogenous RXEG, why there were no enhanced ROS in WT upon treatment of XEG (WT-XEG) in fig 3c?

Response: Compared with EV, XEG1-treatment of the WT or RLP23-expressing *N. benthamiana* does trigger ROS production as shown by the blue line and green line in the fig. 3c, albeit the peak is not that high, with a peak around 500 RLU compared to a peak around 2000 RLU with RXEG1 + XEG1.

7. Fig4a: In both anti-GFP Input and GFP-IP panels, the RXEGL2-eGFP expression is hard to see, so It's unable to conclude the interaction between XEG1 and RXEGL2 present or not.

Response: We agree with the reviewer and adjusted the text as 'For RXEGL2, we could not conclude any association with XEG1 since the protein was poorly expressed.'

8. Fig5a: The RXEG1 (SP+LRR) truncated proteins should be secreted and lost the membrane localization anymore, how will it transduce the signal into cytosol? It may be more informative to elucidate which LRRs are responsible for the association by mutations instead of these truncations.

Response: The rationale behind testing RXEG1^{SP+LRR} truncated protein is to confirm that the association between RXEG1 with XEG1 occurs at the extracellular LRR domain. We would not expect the truncated proteins to be capable of transducing the signal into the cytosol, which was confirmed since RXEG1^{SP+LRR(S)} failed to complement XEG1-induced cell death in TRV:RXEG1-treated *N. benthamiana* (Fig.

2c). We agree with the reviewer that determining which LRRs are responsible for the association with XEG1 is interesting work to do. This analysis, however, is largely dependent on the solved crystal structures. We will perform this analysis in our future work.

9. Fig6 and Supplementary Fig5: BAK1 contributes to the recognition of XEG1. How about its disease susceptibility to Phytophthora?

Response: We performed infection assays on the TRV:BAK1-treated *N. benthamiana* with *Phytophthora parasitica* and we indeed detected increased disease severity and *Phytophthora* biomass when compared with the TRV:GFP-treated plants. This was already shown in Supplementary Figure 9, indicating that BAK1 contributes to *Phytophthora* resistance. This is consistent with the previous report that BAK1 is required for *Phytophthora infestans* resistance in *N. benthamiana* (Chaparro-Garcia et al., 2011 PLoS One).

10. Line101: 1A should be in lower-case.

Response: We thank the reviewer for pointing out this format error and we made the change accordingly.

11. Line 135: It's better to list how many RLPs and RLKs are screened in these 300 LRR genes. Supplementary Fig2a: there is no labeling for RXEG1(S)-eGFP or eGFP expression in the tobacco leaves.

Response: The number of silenced genes was recalculated in a way that the LRR-receptor gene which was tested in different TRV constructs was only count once with the highest silencing efficiency. The silencing of RLP- and RLK-genes is shown in Fig.1 and also in the Supplementary Table 1 where we listed the silencing efficiency according to the TRV construct.

12. No data to show that SOBIR1 is a co-receptor. Modify writing.

Response: We agree with the reviewer and made changes accordingly.

Reviewer #2 (Remarks to the Author):

Review of Wang et al., Nature Communications 2017

The authors present a genome-wide analysis of genes encoding leucine-rich repeat proteins (RLPs) and receptor-like kinases (RLKs) in the model plant species *Nicotiana benthamiana*. The bioinformatics appears to be comprehensive and convincing and resulted in the development of 274 TRV constructs used for gene silencing of all 415

relevant genes. One of the targeted genes, RXEG, is shown to play a role in recognition of a oomycete MAMP (XEG) and, importantly, complementation analysis with a synthetic version of the gene in a RXEG-silenced plant confirmed the gene's involvement in several host responses to XEG. The authors over interpret their data here by saying they have identified the 'receptor' for XEG (on lines 181, line 196 and line 203). In fact they can only conclude at this point that RXEG is required for the host cell death response to the MAMP. They later tone down these statements by saying that RXEG 'potentiates the response to XEG' or 'contributes to plant immunity' (lines 257 and 269).

Response: We agree with the reviewer. We toned down our statement and rephrased the description accordingly.

The evidence for RXEG acting as a receptor for XEG comes in the subsequent section where assays showed that the RLP co-IPs with XEG with some specificity. It does not co-IP with several other MAMPs, including EIX which is bound by tomato LeEIX2 which encodes the tomato ortholog to RXEG. Finally, although silencing of SOBIR only mildly affects the host response to XEG RXEG does associate with the co-receptor SOBIR in a ligand-independent manner. Overall, the experiments are well-conceived, technically good with appropriate controls, and the results are straightforward and convincing.

The paper describes an enormous effort involving bioinformatics, cloning and testing of a large number of TRV constructs, and subsequent assays of the plants with XEG. The results are not especially novel since all of the methods are widely used and there is nothing especially unusual about RXEG (i.e., several other RLPs are known to play a role in plant immunity). However, if the clones are easily available from the authors (this needs to be made clearer) then they constitute a valuable community resource that could greatly impact the PRR field and this paper would be widely cited.

A few minor comments:

Title: Is there a typo here? Should it be receptoromics? Regardless, I would suggest something more descriptive like: Genome-wide analysis of leucine-rich repeat-encoding genes identifies RXEG as the receptor for the fungal glycoside hydrolase 12 protein.

Response: We have changed the title to 'Leucine-rich repeat receptoromics screen reveals that RXEG1 detects glycoside hydrolase 12 MAMPs in *Nicotiana benthamiana*'.

Lines 49-64: A citation to the recently-identified FLS3 gene would be relevant here. Hind, S.R., et al., (2016). Nature Plants 2:16128.

Response: This reference has been added.

Line 77: Change to Solanaceous plant family.

Response: This has been changed accordingly.

Line 77: Provide a citation to the high genomic collinearity of these three species (I am not aware of one).

Response: We made this conclusion based on the genome sequence analyses by the following two references, which we have cited:

1. Bombarely, A. et al. A draft genome sequence of *Nicotiana benthamiana* to enhance molecular plant-microbe biology research. *Molecular Plant-Microbe Interactions* 25, 1523-1530 (2012).

2. Nakasugi, K. et al. De Novo transcriptome sequence assembly and analysis of RNA silencing genes of *Nicotiana benthamiana*. *PLoS One* 8, e59534 (2013).

Line 129: A citation for the VIGS Tool should be provided. Fernandez-Pozo, N., et al., (2015). The SGN VIGS Tool: User-friendly software to design virus-induced gene silencing (VIGS) constructs for functional genomics. *Molecular Plant* 8:486-488.

Response: This reference has been added.

Line 134: I did not see an explicit statement about how the 274 TRV clones will be made available to the community and that is essential for the paper to have serious impact.

Response: We declared the data availability in the manuscript. All the constructs of this study are available from the corresponding author upon reasonable request. We are happy to send the agro-stocks as long as cite this work properly.

Line 137: Some more descriptive language is needed here to define “silencing efficiency”. Presumably they looked at transcript abundance, compared it to a *N. benthamiana* infected with TRV-GFP only and then normalized the data to a control gene known to be stably expressed?

Response: Throughout the manuscript, the transcript levels of LRR receptor genes were monitored by qRT-PCR, normalized with EF-1 α , and expressed as mean fold changes (\pm s.e.m) relative to that in TRV:*GFP*-treated leaves, which was set as 1. This was now stated clearly in the legends showing the silencing efficiency.

Lines 181, 196, 203: Modify all these statements about RXEG being a receptor since that is not supported by the data at this point in the manuscript.

Response: We have rephrased the statements accordingly.

Line 457: The relationship between LeEIX1, LeEIX2 and RXEG needs to be further clarified. Are the two former genes the most closely related to RXEG in the tomato genome? Are they likely orthologs? In this regard I think it would be useful for the authors to provide an amino acid alignment of these three proteins in the supplemental data section.

Response: Amino acid alignment of these three proteins and a phylogenetic tree were shown in the Supplementary Figure 7 to illustrate the relationship of RXEG1 with LeEix1 and LeEix2. The tree shows that RXEG1, RXEGL1 and RXEGL2 are co-orthologs of LeEix1, LeEix2 and another tomato RLP. Thus the three *N. benthamiana* proteins share a common ancestor that is orthologous to the common ancestor of the three tomato proteins. The text has been modified to make this clear.

Line 462: receptors should be co-receptors.

Response: We changed 'receptors' to 'components'.

Reviewer #3 (Remarks to the Author):

In this manuscript, the authors describe a toolkit aiming at silencing genes encoding LRR-containing receptors in *Nicotiana benthamiana*. While no insertion libraries exist to disrupt *N. benthamiana* genes and while only few genetic resources exist for this plant species, establishing a collection of silencing fragments targeting a particular gene family is of particular interest. The author then describe the use of this tool to identify a LRR-receptor-like protein (LRR-RLP) for responses triggered by the *Phytophthora sojae* XEG1 elicitor, which they called RXEG1. They also demonstrate that RXEG1 is able to perceive paralogous elicitors from other *Phytophthora* species and from the fungus *Verticillium dahliae*.

Here are some specific comments on the manuscript, which should help improving this manuscript:

- The authors present an interesting toolkit and also convincing data about the characterisation of RXEG1. However, a detailed phylogenetic analysis is missing as the authors describe for the first time the *Nicotiana benthamiana* LRR-RLKs and LRR-RLPs. The members of the LRR-RLKs family are usually classified in sub-families using the *Arabidopsis* proteins as reference (as in the recent analysis published by Dufayard et al., 2017 doi:10.3389/fpls.2017.00381, for example). However, in the present manuscript, the authors simply define OrthoGroups (based on unknown parameters) and describe "highly homologous" sequences rather than orthologs. Also, because no phylogeny has been constructed, no candidate orthologs of the *N. benthamiana* RXEG1 can be presented nor discussed in this manuscript.

Response: We performed subgroup analysis according to Shiu et al (*Plant Cell*, 2004). These data are present in Table 1, Supplementary Table 2 and Supplementary Figure 1.

Furthermore, indicating that "RXEG1 is highly homologous to LeEIX1" is misleading and

would suggest that LeEIX is the closest ortholog. Also, it would be preferable to perform a classical phylogenetic analysis from which the readers could benefit.

Response: We agree with the reviewer and have made an alignment of these three proteins and a phylogenetic tree to illustrate the relationship of RXEG1 with LeEix1 and LeEix2 as shown in Supplementary Figure 7. The tree shows that RXEG1, RXEGL1 and RXEGL2 are co-orthologs of LeEix1, LeEix2 and another tomato RLP. Thus the three *N. benthamiana* proteins share a common ancestor that is orthologous to the common ancestor of the three tomato proteins. The text has been modified to make this clear.

- The Table S1 is of major importance in this manuscript, thus a particular effort should be given to make it faultless. For example, the column "Full length aa" refers to unknown indications such as LRR, correct or corrected; the number of LRR is sometime missing; one target is of type "TM"; decimals are presented with 9 digits instead a 2 or 3; NbS00001134g0002 appears as sequence under T2, but appears "corrected" under T40; "undet" is missing for plenty of silencing efficiency; NbS00003987g0011, NbS00004687g0019 and NbS00037942g0004 have a "RT primer-R" but no "RT primer-F"; SP are alternatively indicated are lacking, no, no clear; types described as "LRR" would be secreted?

Response: We agree with the reviewer and corrected this table carefully. To be noted, the silencing efficiency was only labeled once the corresponding TRV construct-treated plants were assayed. For certain genes, several different TRV constructs were made and the silencing efficiency may vary depending on the TRV construct.

- While the data presented clearly show that RXEG1 is required for XEG1 responsiveness, it is not possible for the author to state that it is the receptor, unless data showing direct binding of XEG1 are shown (it would be in this context also useful to calculate binding affinity values). Co-immunoprecipitation is not a proof of direct binding. As such, lines 203-204 should, for example, currently rather state "...RXEG1 is required for XEG1-triggered cell death in *N. benthamiana*."

Response: We agree with the reviewer and toned down our statement on RXEG1. We rephrased the sentence as 'Collectively, these results demonstrate that RXEG1 is essential for the XEG1-response in *N. benthamiana*.'

- Title: the current title does not provide much context. I would suggest "Leucine-rich repeat receptoromics analysis reveals the *Nicotiana benthamiana* RXEG1 detects the PAMP glycoside hydrolase 12 protein".

Response: We have changed the title to 'Leucine-rich repeat receptoromics screen

reveals that RXEG1 detects glycoside hydrolase 12 MAMPs in *Nicotiana benthamiana*

- Abstract: line 27: should be "...a complex with the LRR-receptor kinases BAK1 and SOBIR1...".

Response: We have modified the text accordingly.

- It is not entirely correct to refer to SOBIR1 as a co-receptor as it does not directly bind the ligand. Rather, it is proposed that LRR-RLPs form constitutive complexes with SOBIR1 to form what would constitute a bi-partite LRR-RLK, which could then form a ligand-induced complex with the co-receptor BAK1. This should make clearer in the Introduction and the rest of the manuscript. Also, the authors should clearly state that SOBIR1 only plays a role for the function of LRR-RLPs, which is not necessarily clear currently from the Introduction (eg. line 60).

Response: We have modified the statement on SOBIR1 throughout the manuscript and added the text 'the LRR-RLK SOBIR1 functions as an adaptor that associates with multiple LRR-RLPs to form bi-partite equivalents of LRR-RLKs' in the Introduction.

- It is not necessarily appropriate to name NbS00035240g0005 as RXEGL2 given that it has only 12 LRRs compared to 26 for RXEG1 and RXEGL1.

Response: NbS00035240g0005 is one of the closest homologs of RXEG1 (Supplementary Figure 7) and we name it as RXEGL2 for short.

- Decide whether you'd like to use the MAMP or PAMP abbreviation. At the moment, both abbreviations are used throughout the manuscript.

Response: Thanks the reviewer for pointing out this inconsistency. We chose to use MAMP throughout the manuscript.

- Fig 1c: For a better reading, (and using a clock analogy) the values should start with 50 at 12:00 rather than at 10:00.

Response: This figure has been changed accordingly.

- Fig 3a, 3c, 6d, S4a: in the panels, please replace "TRV:GFP-EV" with "TRV:GFP + EV", etc ...

Response: These have been changed accordingly.

- Fig S1b: Percentages should be followed by the ID of the corresponding genes. Also the corresponding legend is missing for this figure panel.

Response: We modified the figure accordingly and added the corresponding figure legend.

- Fig S6: In the fifth line, RXEGL1 should be indicated rather than RXEG1. It is unclear what the differences are between SP+LRR and SP+LRR(s). For RXEGL1SP+TM-tail, 910 may be missing to indicate that the blue part is the eJM.

Response: We agree with the reviewer and made changes accordingly. For SP+LRR and SP+LRR^(s), the protein sequences are the same, but the nucleotide sequences are different. The sequences are shown in Supplementary Note 1.

- Please indicate which HMM models have been used for LRR and TM.

Response: The HMM model for the LRR domain is PFAM ID: PF00560.31; downloaded from <http://pfam.xfam.org>. The TM domain was predicted via the web tool TMHMM Server V. 2.0 (<http://www.cbs.dtu.dk/services/TMHMM>). This information was added in the Methods section.

- Please list and describe all the elicitors used in the manuscript (nlp20, EIX, Ps119627, Ps138787, Avh241, ...) in the Methods section.

Response: We added these descriptions in the Methods section.

- It is unclear how 1 μ M XEG1 is obtained. Is GenScript supplying a peptide, protein or gene? Is the protein tagged?

Response: XEG1 is a purified protein supplied by Genscript. A C-terminal His-tag was included for purification of the protein. We mentioned this information in the Methods section.

- line 36-37: since all MAMPs are "elicitors released from microbes", the sentence should be rephrased.

Response: We have changed 'elicitors' to 'proteins'.

- line 39: "plethora" should be replaced with a more accurate range of gene numbers.

Response: We replaced "plethora" with "hundreds".

- line 43: Please rephrase "divergent in their ectodomain".

Response: This has been rephrased as 'Cell surface receptors have diverse ectodomains'

- line 65: should be "sequences microbial genomes".

Response: This has been changed.

- line 80: Please replace "Multiple PAMPs and elicitors" with "Multiple PAMPs including elicitors".

Response: This has been changed.

- line 99-101: The sentence need to be rephrased. It is unclear why 4 LRR-Ser/Thr kinase needed corrections.

Response: All the predicted LRR-containing proteins were retrieved after blast searches and were corrected according to the available RNA-seq data. Of these, sequences which were predicted to encode LRR-serine/threonine kinases lacking a clear transmembrane domain were also corrected. In the submitted manuscript, four LRR-STKs were labeled as corrected to be LRR-RLKs. After a close check of the gene models, we found that only NbS00023516g0009 and NbS00016534g0006, but not the other two, encode full length LRR-RLKs with a clear transmembrane domain. This has been corrected in the manuscript.

- Line 103: no results are presented to support the existence of “two copies of highly homologous LRR receptors”.

Response: A phylogenetic tree was added in the Supplementary Figure 1 to illustrate the relationships of the identified LRR receptors.

- line 135: Please replace “Table 1” with “Table S1”.

Response: This has been changed accordingly.

- line 148: should be “Phylogenetic analysis”.

Response: This has been changed accordingly.

- line 175 and 193: should be “infiltrated” rather than “treated”.

Response: This has been changed accordingly.

- line 182: Please rephrase as the target genes could also encode regulatory LRR-RLK/P.

Response: We have rephrased the sentence as ‘These results indicated that the TRV:RXEG1-1 and TRV:RXEG1-2 targeted gene(s) encode a component essential for XEG1 recognition.’

- line 197: The two sentences should be merged.

Response: This has been changed.

- line 203: Please rephrase the sentence as RXEG1 requirement doesn’t demonstrate its role as bona fide receptor.

Response: This has been rephrased as ‘Collectively, these results demonstrate that RXEG1 is essential for the XEG1-response in *N. benthamiana*.’

- lines 244-245: rewrite to “...and confers disease resistance”.

Response: This has been changed.

- line 255-257: should be “over-expression”.

Response: This has been changed.

- line 269: should be “to confer resistance to Phytophthora when expressed in *N. benthamiana*”.

Response: This has been changed.

- line 297: As RXEGL2-eGFP is not detected after GFP-IP in Fig 4a. Also, it can't be stated that this receptor doesn't interact with XEG1-HA.

Response: We have rephrased this part as ‘As shown in Fig. 4a, we repeatedly detected specific association between XEG1 and RXEG1, but not between XEG1 and either of RXEGL1 or RLP23. For RXEGL2, we could not conclude any association with XEG1 since the protein was poorly expressed.’

- line 301: Please rephrase, as it is confusing to present the tomato homologs of RXEG1 as no phylogenetic analyses have been performed.

Response: The sequence alignment of these three proteins and a phylogenetic tree were shown in the Supplementary Figure 7 to illustrate the relationship between RXEG1 and tomato orthologs.

- line 307: There is no dependent relation between EIX not interacting with RXEG1 and EIX not being perceived in *N. benthamiana*. Please rephrase.

Response: We have deleted the sentence ‘consistent with our previous observation that *N. benthamiana* does not respond to the elicitor EIX’.

- line 329: It may be indicated that SP-TM-tail is in fact SP-eJM-TM-tail.

Response: We agree with the reviewer and referred ‘SP-eJM-TM-tail’ to ‘ Δ LRR’ for short.

- lines 356 and 486: see comment above about SOBIR1 and the term co-receptor.

Response: This has been changed.

- line 368: should be “associated with LRR-type PRRs”.

Response: This has been changed.

- line 432: Please replace “LRR receptor” with “LRR transmembrane receptor”.

Response: We have rephrased this sentence as ‘we generated 274 TRV-based constructs to silence all of the 412 identified genes that encode predicted membrane-localized LRR receptors in *N. benthamiana*.’

- line 458: Please replace “receptors function” with “receptors whose function”.

Response: This has been changed.

- lines 458-463: it should be note here that another LRR-RLP, CSPR, was also previously shown to associate with SOBIR1 but not depend on it for its function (Saur et al., 2016).

Response: This information has been added in the text as 'One exception is NbCSPR, which constitutively associates with SOBIR1 but does not require SOBIR1 for its function in csp22 responses'.

- line 467: "variable" may be used instead of "controversial". Also, it may be worthwhile here to mention that these results might suggest that other SERK proteins are preferentially used by these receptors.

Response: This has been changed accordingly.

- line 497: Please replace "for those are" with "for those which are".

Response: This has been changed.

- line 507: Please indicate which versions of the genome releases were used for sequence analysis.

Response: This has been added in Methods section.

- line 529: References for expressions vectors are missing.

Response: This has been added.

- line 531: The reference to EIX sequence is presented as Rotblat et al., 2002. However, the TvEIX sequence was rather described in Furman-Matarasso et al., 1999.

Response: This has been changed accordingly.

Reviewers' Comments:

Reviewer #1 (Remarks to the Author):

The authors have addressed my concerns. It's a piece of nice work. Just some minor edits for writing.

1. Title:

I would suggest the authors to re-consider a title to precisely capture what they have done. The currently revised title has grammar mistakes and is confusing. Considering that 1) the authors do not have data to show direct binding, 2) members of LRR-RLKs or RLPs may not be receptors rather signaling components, I would suggest to a title

Genome-wide analysis of leucine-rich repeat receptor-like genes reveals that RXEG is required for the fungal glycoside hydrolase 12 protein-triggered responses.

2. To avoid confusing and misleading, I suggest the authors not to use receptoromics throughout the ms.

3. change "LRR receptor genes" to "LRR receptor-like genes/proteins" throughout the ms.

3. Abstract

Plant genomes encode hundreds of such receptors

Change to "such receptor-like proteins"

Reviewer #2 (Remarks to the Author):

The changes I requested have been satisfactorily addressed and I believe the paper now merits publication.

Reviewer #3 (Remarks to the Author):

In general, I am happy with this revised manuscript, which adequately addresses my previous comments.

There are however still a few points that should be addressed, as listed below:

- In the manuscript, the authors are using gene IDs corresponding to the *Nicotiana benthamiana* genome release v0.4.4, while a new version, v1.0.1. is available since March 2015. Also, it is unclear at which point the "protein and cDNA sequences" were further validated. As the gene IDs presented are outdated, it would be preferable to present the actual Sol Genomics IDs (the

correspondence can be found here: ftp://ftp.solgenomics.net/genomes/Nicotiana_benthiana/annotation/Niben044vsNiben101_GeneID_mapping.txt) or to present the QUT benthgenome IDs which are supported by RNAseq data.

- The authors should also check if their protein models are still correct, and they may also identify signal peptides which were missing for many protein models. Many kinase domains are also unusually short (<250 aa) in the RLKs presented; this may have been corrected in the latest models. As the authors indicate that "All these sequences [...] have been deposited in Genbank", it would be preferable that the quality of these gene models is at least equal to the latest release.

- For the phylogenetic analysis of the LRR-RLPs presented in the new Figure S7b, it is unclear why only 15 out of 81 LRR-RLPs are used in the phylogenetic analysis. Also, it still can't be stated which NbLRR-RLPs is the closest ortholog of LeEIX1/2. Furthermore, the method of phylogenetic analysis is not described (the Materials and Methods section describe a phylogenetic analysis based on kinase domains which is not relevant for this LRR-RLP analysis). An elegant method for phylogenetic analysis of LRR-RLPs would be to use the C3-D domains as it previously used in Petre et al., 2014 (doi: 10.3389/fpls.2014.00111).

- Fig. 6b,c,e require an Anova statistical analysis as it is otherwise not possible to make any statement about potential differences between silencing of BAK1 and SOBIR1.

- Also, in Fig. 6f, it is not clear why the anti-mRFP was spliced (as indicated by the bar, I suppose). Does this mean that this blot (or part of it) is coming from different biological experiments?

- Instead of the current Fig S8, it would be informative to show the RXEG1 sequence with the different domain aligned and annotated as previously done in Zipfel et al., 2006 (doi: 10.1016/j.cell.2006.03.037) or Jehle et al., 2013 (doi: 10.1105/tpc.113.110833), for example. This would illustrate that RXEG1 has actually 31 LRRs instead of the reported 26 LRRs, with an island domain between LRR27 and LRR28.

- The description of the new concept RLFP (RLK+RLP) is rather vague. Would the secreted LRRs from the PGIP family also be considered as LRR-RLFPs? Would OsCEBiP, which is GPI-anchored, also be considered as a LysM-RLFP? Also, LRR receptor may still be preferred to LRR-RLFP. There is frankly no need to bring up yet another acronym, and I would suggest that the authors simply refer to LRR-RLKs and LRR-RLPs.

- In Tab S1, RT primer-F are still missing for T13 and T14, and construct primer-R are still missing for T43 and T210.

- line 50: "flagellin receptors"

- line 51: "elongation factor Tu"

- The sentence line 50-57 need to be rephrased as the sentence lists documented PRRs but then switch to PAMPs.

- line 57: "..., and fungal endopolygalacturonases, respectively"
- line 59: "LRR-RLP CSPR and the LRR-RLK CORE"
- line 60-61: "by forming complexes with multiple LRR-type PRRs"
- line 72-73: it is not the "function" of XEG1 to be recognized as a MAMP in the apoplast; please rephrase.
- At line 104, it is unclear why the authors mention "NbS00023516g0009 and NbS00016534g0006 which were annotated as LRR-serine/threonine kinases before correction" while many other LRR-RLKs have also been corrected.
- Line 129: please rephrase "by blasting sequences".
- Line 219-220: please rephrase "Infection of leaves silencing of SOBIR1 was used as a positive control."
- Line 220: please rephrase "those"
- The URL indicated line 418 is not correct (an underscore is missing) and should be replaced with http://sydney.edu.au/science/molecular_bioscience/sites/benthamia na/

Response to Reviewers' comments:

Reviewer #1 (Remarks to the Author):

The authors have addressed my concerns. It's a piece of nice work. Just some minor edits for writing.

Response: Thanks for your constructive suggestions for this manuscript.

1. Title:

I would suggest the authors to re-consider a title to precisely capture what they have done. The currently revised title has grammar mistakes and is confusing. Considering that 1) the authors do not have data to show direct binding, 2) members of LRR-RLKs or RLPs may not be receptors rather signaling components, I would suggest to a title
Genome-wide analysis of leucine-rich repeat receptor-like genes reveals that RXEG is required for the fungal glycoside hydrolase 12 protein-triggered responses.

Response: We agree with the reviewer and rephrased the title as 'Leucine-rich repeat receptor-like gene screen reveals that *Nicotiana* RXEG1 regulates glycoside hydrolase 12 MAMP detection' to meet the 15-word limitation in the title.

2. To avoid confusing and misleading, I suggest the authors not to use receptoromics throughout the ms.

Response: We deleted the word 'receptoromics' throughout the manuscript.

3. change "LRR receptor genes" to "LRR receptor-like genes/proteins" throughout the ms.

Response: This has been changed accordingly.

4. Abstract

Plant genomes encode hundreds of such receptors

Change to "such receptor-like proteins"

Response: This has been rephrased to 'Plant genomes contain hundreds of such receptor-like genes'.

Reviewer #2 (Remarks to the Author):

The changes I requested have been satisfactorily addressed and I believe the paper now merits publication.

Response: Thanks for your constructive suggestions for this manuscript.

Reviewer #3 (Remarks to the Author):

In general, I am happy with this revised manuscript, which adequately addresses my previous comments.

Response: Thanks for your constructive suggestions for this manuscript.

There are however still a few points that should be addressed, as listed below:

- In the manuscript, the authors are using gene IDs corresponding to the *Nicotiana benthamiana* genome release v0.4.4, while a new version, v1.0.1. is available since March 2015. Also, it is unclear at which point the "protein and cDNA sequences" were further validated. As the gene IDs presented are outdated, it would be preferable to present the actual Sol Genomics IDs (the correspondence can be found here: ftp://ftp.solgenomics.net/genomes/Nicotiana_benthamiana/annotation/Niben044vsNiben101_GeneID_mapping.txt) or to present the QUT benthgenome IDs which are supported by RNAseq data.

Response: We agree with the reviewer and performed analyses to search for LRR-RLPs and LRR-RLKs using the latest version (v1.0.1) of *Nicotiana benthamiana* genome and mapped all the targeted genes in the genome version v1.0.1 to each TRV construct according to the cloned silencing fragments as shown in the Supplementary Table 1.

- The authors should also check if their protein models are still correct, and they may also identify signal peptides which were missing for many protein models. Many kinase domains are also unusually short (<250 aa) in the RLKs presented; this may have been corrected in the latest models. As the authors indicate that "All these sequences [...] have been deposited in Genbank", it would be preferable that the quality of these gene models is at least equal to the latest release.

Response: We re-checked all the gene models using the available RNA-seq data via the website <http://benthgenome.qut.edu.au/> and made corrections accordingly.

Only the gene models of RXEG1 (Niben101Scf03925g01010.1/genbank ID: MG010652), RXEGL1 (Niben101Scf00975g01015.1/genbank ID: MG010653) and RXEGL2 (Niben101Scf03925g02017.1/genbank ID: MG010654) are deposited in Genbank. The rest gene models listed in the Supplementary Table 1 were not accepted by Genbank since they are not verified experimentally.

- For the phylogenetic analysis of the LRR-RLPs presented in the new Figure S7b, it is unclear why only 15 out of 81 LRR-RLPs are used in the phylogenetic analysis. Also, it can't be stated which NbLRR-RLPs is the closest ortholog of LeEIX1/2. Furthermore, the method of phylogenetic analysis is not described (the Materials and Methods section describe a phylogenetic analysis based on kinase domains which is not relevant for this LRR-RLP analysis). An elegant method for phylogenetic analysis of LRR-RLPs would be to use the C3-D domains as it previously used in Petre et al., 2014 (doi: 10.3389/fpls.2014.00111).

Response: A phylogenetic tree was built using LeEix1/2 and all the identified *N. benthamiana* RLPs containing the C3-D domain following the method described by Fritz-Laylin et al. 2005 and Petre et al. 2014 as shown in the Supplementary Figure 8b.

- Fig. 6b,c,e require an Anova statistical analysis as it is otherwise not possible to make

any statement about potential differences between silencing of BAK1 and SOBIR1.

Response: One-way ANOVA statistical analysis was performed and shown in Fig. 6c and 6e. We did not compare the silencing efficiency of TRV:BAK1 and TRV:SOBIR1 in Fig.6b since plants treated with either construct always get around 80-90% silencing efficiency. These slight changes in silencing efficiency did not make significant difference in plant response to XEG1 treatment.

- Also, in Fig. 6f, it is not clear why the anti-mRFP was spliced (as indicated by the bar, I suppose). Does this mean that this blot (or part of it) is coming from different biological experiments?

Response: In Fig. 6f, we showed the same blot detected by anti-mRFP and used the bar to distinguish the two parts visualized in different contrasts. Due to the weak cell death caused by expression of BAK1, we always detected less protein (BAK1, SOBIR1 and RXEG1) abundances when co-express RXEG1 with both BAK1 and SOBIR1 than co-express RXEG1 with either BAK1 or SOBIR1 in *Nicotiana benthamiana*. Therefore, we showed the anti-mRFP detected BAK1 in samples co-expressing RXEG1 with both BAK1 and SOBIR1 in higher contrast to make the bands easier to visualize.

- Instead of the current Fig S8, it would be informative to show the RXEG1 sequence with the different domain aligned and annotated as previously done in Zipfel et al., 2006 (doi: 10.1016/j.cell.2006.03.037) or Jehle et al., 2013 (doi: 10.1105/tpc.113.110833), for example. This would illustrate that RXEG1 has actually 31 LRRs instead of the reported 26 LRRs, with an island domain between LRR27 and LRR28.

Response: We recalculated the LRR composition of RXEG1 following the description of Zipfel et al., 2006 and Jehle et al., 2013 and added this data in the Supplementary Figure 3.

- The description of the new concept RLFP (RLK+RLP) is rather vague. Would the secreted LRRs from the PGIP family also be considered as LRR-RLFPs? Would OsCEBiP, which is GPI-anchored, also be considered as a LysM-RLFP? Also, LRR receptor may still be preferred to LRR-RLFP. There is frankly no need to bring up yet another acronym, and I would suggest that the authors simply refer to LRR-RLKs and LRR-RLPs.

Response: This has been changed throughout the manuscript.

- In Tab S1, RT primer-F are still missing for T13 and T14, and construct primer-R are still missing for T43 and T210.

Response: These primers were added accordingly.

- line 50: "flagellin receptors"

Response: This has been changed.

- line 51: "elongation factor Tu"

Response: This has been changed.

- The sentence line 50-57 need to be rephrased as the sentence lists documented PRRs but then switch to PAMPs.

Response: This has been rephrased.

- line 57: "..., and fungal endopolygalacturonases, respectively"

Response: This has been changed.

- line 59: "LRR-RLP CSPR and the LRR-RLK CORE"

Response: This has been changed.

- line 60-61: "by forming complexes with multiple LRR-type PRRs"

Response: This has been changed.

- line 72-73: it is not the "function" of XEG1 to be recognized as a MAMP in the apoplast; please rephrase.

Response: This has been rephrased as 'The glycoside hydrolase 12 (GH12) protein XEG1 identified from the soybean root rot pathogen *Phytophthora sojae* is recognized in plant apoplast as a novel MAMP'

- At line 104, it is unclear why the authors mention "NbS00023516g0009 and NbS00016534g0006 which were annotated as LRR-serine/threonine kinases before correction" while many other LRR-RLKs have also been corrected.

Response: This description is not applicable for receptors identified in the *N. benthamiana* version genome v1.0.1 and has been removed from the manuscript.

- Line 129: please rephrase "by blasting sequences".

Response: This has been rephrased as 'The silencing specificity was determined by Blast analysis using the VIGS tool in the Sol Genomic Network (SGN) website'

- Line 219-220: please rephrase "Infection of leaves silencing of SOBIR1 was used as a positive control."

Response: This has been rephrased as 'Leaves silencing of *SOBIR1* were inoculated and used as a positive control.'

- Line 220: please rephrase "those"

Response: 'those' has been changed to 'leaves'.

- The URL indicated line 418 is not correct (an underscore is missing) and should be replaced with http://sydney.edu.au/science/molecular_bioscience/sites/benthamiana/

Response: Thanks for pointing out this error. The latest URL <http://benthgenome.qut.edu.au/> was added.

Reviewers' Comments:

Reviewer #3 (Remarks to the Author):

The authors have done a great job in addressing the previous comments.

Below are just a few remaining minor comments:

- Line 35, should read Fig. 1, not Fig. 1c.
- Line 138 to 143 refers to silencing constructs which are different from the one presented Fig S2a. T106 and T211 are presented in the text, but T117 and T227 are shown in Fig S2a. Furthermore, T140, T123, T136, and T211 appear to target 2 genes each, but the text refer to six targeted genes. This has to be clarified.
- Line 143, Fig. 1a should read Fig. 2a.
- Line 337, remove “and this domain”.
- Line 506, the XEG1 purification method need to be described.
- In Tab S2, a value is missing in cell F4.
- In Fig S1, it is impossible to read the gene names.
- In Fig S9, RXEGL1 should replace the second RXEG1.
- Nlp20 description is missing. Please confirm that the *Phytophthora parasitica* variant is used.

Response to reviewer's comments:

Reviewer #3:

The authors have done a great job in addressing the previous comments.

Response: We thank you for your suggestions to improve our manuscript.

Below are just a few remaining minor comments:

- Line 135, should read Fig. 1, not Fig. 1c.

Response: We thank the reviewer for pointing out this error and we made the correction accordingly.

- Line 138 to 143 refers to silencing constructs which are different from the one presented Fig S2a. T106 and T211 are presented in the text, but T117 and T227 are shown in Fig S2a. Furthermore, T140, T123, T136, and T211 appear to target 2 genes each, but the text refer to six targeted genes. This has to be clarified.

Response: We thank the reviewer for pointing out this error and we made corrections accordingly. The description about the four silencing constructs were rephrased as “Collectively, these four constructs target two genes encoding LRR-RLPs and six genes encoding LRR-RLKs with less than ten extracellular LRRs. Only the two genes targeted by T136 encode an RD kinase domain while the remaining four LRR-RLK genes encode non-RD kinases with a GN or GH substitution (Supplementary Fig. 2b).”

- Line 143, Fig. 1a should read Fig. 2a.

Response: We thank the reviewer for pointing out this error and we made the correction accordingly.

- Line 337, remove “and this domain”.

Response: We removed “and this domain”.

- Line 506, the XEG1 purification method need to be described.

Response: This has been added in the Methods section.

- In Tab S2, a value is missing in cell F4.

Response: “RD” was added in the cell F4 in the Supplementary Table 2.

- In Fig S1, it is impossible to read the gene names.

Response: We added the figure with high resolution in the Supplementary Fig. 1.

- In Fig S9, RXEGL1 should replace the second RXEG1.

Response: This has been changed.

- Nlp20 description is missing. Please confirm that the *Phytophthora parasitica* variant is used.

Response: nlp20 used in this study is indeed from *Phytophthora parasitica* NLP and this information was added in the text as “The interaction between RLP23 and BAK1 was detected upon treatment with the ligand nlp20 (PpNLP),...”.